# Thermally stable single atom Pt/m-Al$_2$O$_3$ for selective hydrogenation and CO oxidation

Zailei Zhang[1], Yihan Zhu[2], Hiroyuki Asakura[3,4], Bin Zhang[1], Jiaguang Zhang[1], Maoxiang Zhou[5], Yu Han[2,6], Tsunehiro Tanaka[3,4], Aiqin Wang[5], Tao Zhang[5] & Ning Yan[1]

Single-atom metal catalysts offer a promising way to utilize precious noble metal elements more effectively, provided that they are catalytically active and sufficiently stable. Herein, we report a synthetic strategy for Pt single-atom catalysts with outstanding stability in several reactions under demanding conditions. The Pt atoms are firmly anchored in the internal surface of mesoporous Al$_2$O$_3$, likely stabilized by coordinatively unsaturated pentahedral Al$^{3+}$ centres. The catalyst keeps its structural integrity and excellent performance for the selective hydrogenation of 1,3-butadiene after exposure to a reductive atmosphere at 200 °C for 24 h. Compared to commercial Pt nanoparticle catalyst on Al$_2$O$_3$ and control samples, this system exhibits significantly enhanced stability and performance for n-hexane hydro-reforming at 550 °C for 48 h, although agglomeration of Pt single-atoms into clusters is observed after reaction. In CO oxidation, the Pt single-atom identity was fully maintained after 60 cycles between 100 and 400 °C over a one-month period.

[1] Department of Chemical and Biomolecular Engineering, National University of Singapore, 4 Engineering Drive 4, 117585 Singapore, Singapore. [2] Advanced Membranes and Porous Materials Center, Physical Sciences and Engineering Division, King Abdullah University of Science and Technology, Thuwal 23955-6900, Saudi Arabia. [3] Department of Molecular Engineering, Graduate School of Engineering, Kyoto University, Kyoto 615-8510, Japan. [4] Elements Strategy Initiative for Catalysts & Batteries (ESICB), Kyoto University, Kyoto 615-8245, Japan. [5] State Key Laboratory of Catalysis, Dalian Institute of Chemical Physics, Chinese Academy of Sciences, Dalian 116023, China. [6] KAUST Catalysis Center, Physical Sciences and Engineering Division, King Abdullah University of Science and Technology, Thuwal 23955-6900, Saudi Arabia. Correspondence and requests for materials should be addressed to N.Y. (email: ning.yan@nus.edu.sg).

Element sustainability has become a global issue. Maintaining current consumption rate, there are 22 elements facing depletion within the coming 50 years, including almost all Platinum Group metals that are crucial catalyst components[1]. Despite the critical role of Pt in catalysis[2], it is rare, in short supply in recent years and has no adequate alternatives. In this regard, single-atom catalysts (or atomically dispersed catalysts), in which all the metal atoms are exposed on the support available for catalytic reactions, could help to address the problem[3–7]. The electronic properties of isolated metal atoms may be distinctly different from the active sites in bulk materials and nanoparticles, potentially triggering innovative applications and enabling more effective usage of noble metal elements[8–28].

A challenge in the development of single-atom catalysts is the increasing difficulty to stabilize single-atom species under drastic reaction conditions. Many industrially important catalytic processes involving Pt catalysts, such as reforming of hydro-carbons in petroleum refineries, are operated at several hundred degrees under oxidative or reductive atmosphere[29,30]. In this context, single-atom catalysts that are able to withstand harsh reaction conditions are highly desirable. Very recently, Datye and co-workers have illustrated the trapping of atomic Pt species leached from alumina onto the defect sites in rod or polyhedral shaped ceria at 800 °C (ref. 31). Corma *et al.* reported subnanometric Pt species finely confined within the internal framework cavities of MCM-22, which withstands thermal treatment in air up to 540 °C, and is more stable in propane dehydrogenation than catalysts prepared via the conventional wet-impregnation method[32]. Flytzani-Stephanopoulos and Sykes *et al.* developed single-atom alloy catalysts, in which isolated Pt atoms substituted into the Cu(111) surface, that are stable for butadiene hydrogenation at 160 °C for more than 46 h (ref. 33). These examples highlight the importance of manipulating the interactions between metal atoms and the host support to achieve high stability without losing reactivity in single-atom catalysis.

Herein, we report a highly stable, atomically dispersed Pt catalyst supported on mesoporous $Al_2O_3$. The Pt − Al − O system is chosen because $Al_2O_3$ is a common support for Pt in industrial and environmental applications. The catalyst was prepared by a modified sol-gel solvent vaporization self-assembly method[34], followed by calcination in air and reduction with $H_2$. Encouragingly, the catalyst maintained the catalytic activity as well as structural integrity when scrutinized in a series of reactions under drastic conditions with long durations.

## Results

**Catalyst synthesis and characterization.** In the first step, $C_9H_{21}AlO_3$, $(EO)_{20}(PO)_{70}(EO)_{20}$ triblock copolymer (P123) and $H_2PtCl_6$ were mixed and dissolved in ethanol. With continued evaporation of the solvent, the amphiphilic P123 macromolecules associated with $C_9H_{21}AlO_3$ spontaneously assembled into a highly ordered, hexagonally arranged mesoporous structure with Pt precursor encapsulated in the matrix. The resulting gel was calcined in air at 400 °C, during which the P123 template decomposed while the $C_9H_{21}AlO_3$ was transformed into a rigid, well-aligned mesoporous $Al_2O_3$ framework (m-$Al_2O_3$, as shown in Fig. 1). The reductive behaviour of Pt/m-$Al_2O_3$-$O_2$ over $H_2$ was determined from their temperature-programmed reduction (TPR) profiles (Supplementary Fig. 1). A single reduction peak located at approximately 400 °C was observed for the 0.2 wt% Pt loading sample, while two peaks appeared for the 0.5 and 2.0 wt% Pt loading samples. Based on TPR profiles, 400 °C was selected as the unified reducing temperature. After reduction, ICP-OES analysis indicates the Pt loading of the samples to be 0.20, 0.48 and 1.93 wt%, respectively, in excellent agreement with the amount of the Pt precursor used (Supplementary Table 1).

The pre-catalysts and catalysts were labelled in the following manner: Pt weight percentage, nature of the support (for example, m-$Al_2O_3$, referring to mesoporous $Al_2O_3$), and treatment atmosphere. For instance, 0.2Pt/m-$Al_2O_3$-$O_2$ refers to 0.2 wt% Pt supported on mesoporous $Al_2O_3$ which was calcined in air at 400 °C without reduction, whereas 0.5Pt/m-$Al_2O_3$-$H_2$ refers to 0.5 wt% of Pt on mesoporous $Al_2O_3$, consecutively calcined in air and reduced with $H_2$ at 400 °C. FT-IR spectra (Supplementary Fig. 2) suggest complete removal of the P123 triblock copolymer and the ligand of aluminium isopropoxide in all samples. Supplementary Fig. 3 provides a photo of the aforementioned samples: 0.2Pt/m-$Al_2O_3$-$O_2$ and 0.2Pt/m-$Al_2O_3$-$H_2$ are grey, while the others are black. This contrasts with pure m-$Al_2O_3$ which is light yellow. Organic elemental analysis suggested the presence of nitrogen (0.97 wt%) and carbon (0.79 wt%) in 0.2Pt/m-$Al_2O_3$-$O_2$ whereas the content of these elements became negligible in 0.2Pt/m-$Al_2O_3$-$H_2$ (Supplementary Table 2).

The samples were subsequently investigated by scanning transmission electron microscopy (STEM) with high-angle annular dark-field (HAADF), transmission electron microscopy (TEM), Brunauer–Emmett–Teller (BET) surface area analysis, $H_2$-$O_2$ and CO titration, diffuse reflectance infrared Fourier transform spectroscopy (DRIFTS), $^{27}$Al magic-angle spinning nuclear magnetic resonance spectroscopy (MAS NMR), X-ray diffraction (XRD) and X-ray absorption spectroscopy (XAS) to probe the characteristics of the support and the Pt species. These materials exhibit similar surface areas (200.6 to 227.3 $m^2 g^{-1}$), pore volumes (0.51 to 0.56 $ml g^{-1}$), and a diffraction peak at around 0.8° in the small-angle XRD pattern, unambiguously proving the mesoporous structure (Supplementary Figs 4 and 5, Supplementary Table 3). No Pt peaks are detected from XRD patterns in 0.2Pt/m-$Al_2O_3$-$O_2$, 0.2Pt/m-$Al_2O_3$-$H_2$, 0.5Pt/m-$Al_2O_3$-$O_2$ and 0.5Pt/m-$Al_2O_3$-$H_2$, suggesting that most Pt species are present in small nanoclusters and/or isolated atoms. Weak diffraction peaks for Pt are found on 2.0Pt/m-$Al_2O_3$-$O_2$ and 2.0Pt/m-$Al_2O_3$-$H_2$, indicating the formation of Pt nano-particles (Supplementary Fig. 6).

Consistent with BET and XRD analysis, the TEM images of 0.2Pt/m-$Al_2O_3$-$O_2$ and 0.2Pt/m-$Al_2O_3$-$H_2$ show well-aligned, mesoporous $Al_2O_3$ without any Pt nanoparticles (Fig. 2a,b, Supplementary Fig. 7a–d), and TEM-EDS images reveal that Pt is uniformly dispersed throughout the sample (Fig. 2c, Supplementary Fig. 7e). Indeed, HAADF-STEM technique clearly demonstrates the existence of isolated Pt atoms with high number density in 0.2Pt/m-$Al_2O_3$-$H_2$ (Fig. 2d) and 0.2Pt/m-$Al_2O_3$-$O_2$ (Fig. 2f). This is remarkable, since 0.2Pt/m-$Al_2O_3$-$H_2$ was even treated at 400 °C in the presence of $H_2$. Apart from the dominant amount of isolated atoms, a few small clusters that exhibit considerable structural dynamics under electron beam were also observed. A snapshot image (Fig. 2g,h) shows that the cluster has loosely packed atoms with the interatomic distances longer than those observed in metallic Pt[35]. However, it is difficult to affirm whether these small clusters are formed by loose packing of single Pt atoms, or by electron beam-induced fragmentation of the close packed structure. More images and intensity profiles of Pt species in these samples can be found in Supplementary Figs 8 and 9. The numbers of atoms for the nearest neighbour distance from 525 single atoms identified on 0.2Pt/m-$Al_2O_3$-$O_2$ and 532 single atoms identified on 0.2Pt/m-$Al_2O_3$-$H_2$ were analysed (Fig. 2e, Supplementary Fig. 9c), showing that a vast majority of Pt atoms are well separated from each other (distance $\geq$ 0.5 nm). Not surprisingly, Pt nanoclusters and nanoparticles are observed in samples with 0.5 and 2.0 wt% Pt loading. In addition, the enhanced Pt content is detrimental to the formation of high quality hexagonally arranged mesoporous $Al_2O_3$ (Supplementary Figs 10 and 11).

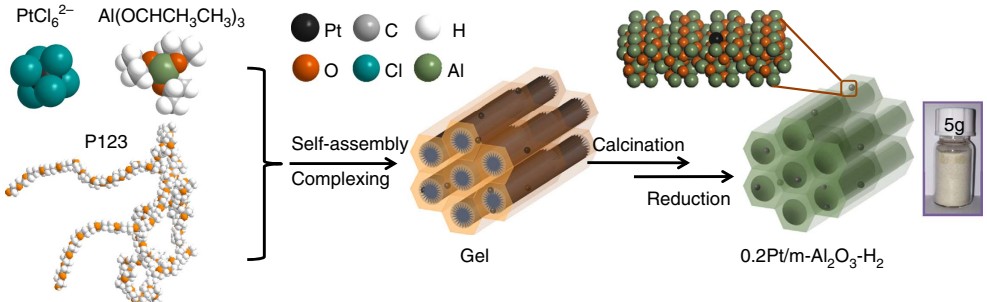

**Figure 1 | Schematic illustration of the 0.2Pt/m-Al$_2$O$_3$-H$_2$ synthesis process.** Aluminum isopropoxide, P123, and H$_2$PtCl$_6$ mixture ethanolic solution self-assembled into a gel after ethanol evaporation at 60 °C. The gel was calcined at 400 °C and reduced in 5% H$_2$/N$_2$ at 400 °C, forming the single atom catalyst 0.2Pt/m-Al$_2$O$_3$-H$_2$.

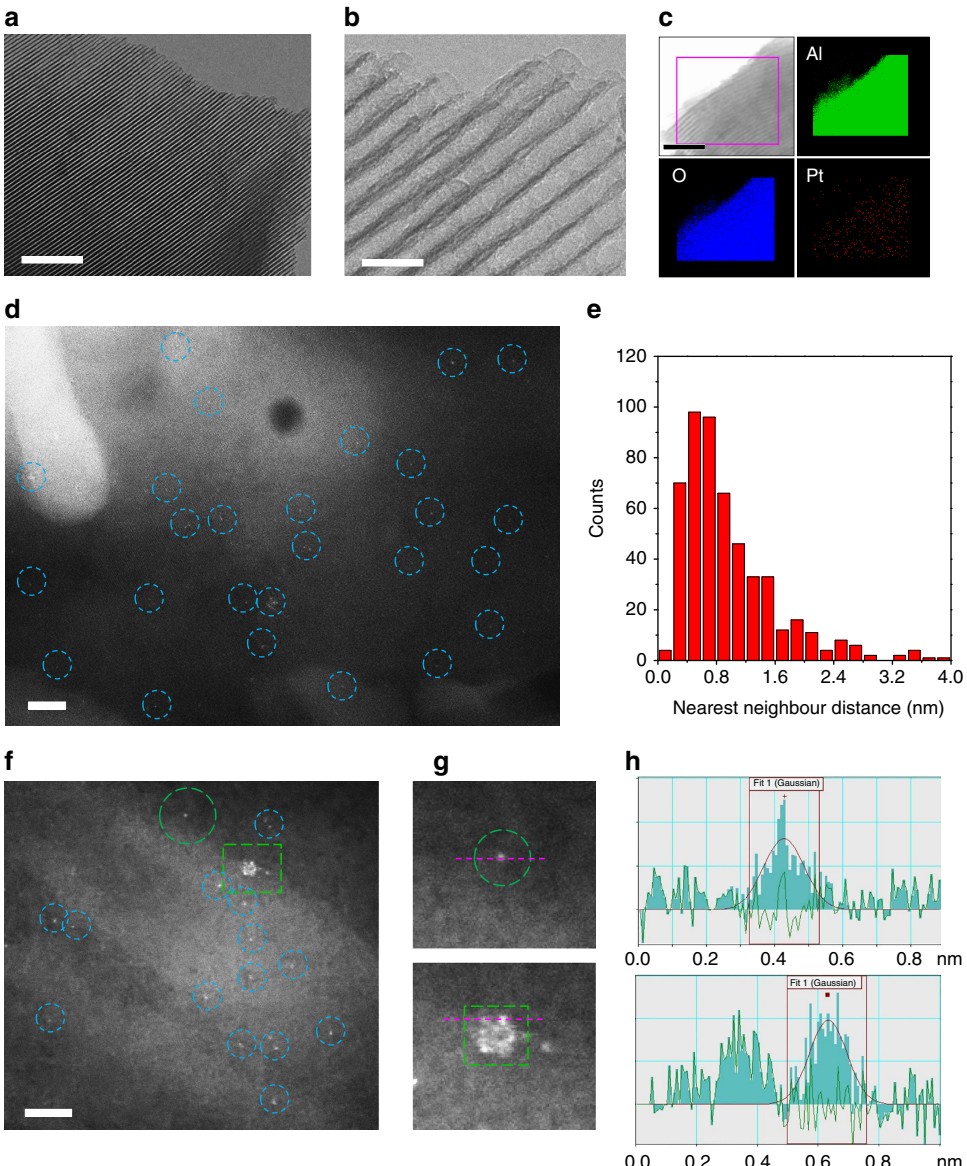

**Figure 2 | Characterization of single-atom Pt materials.** For 0.2Pt/m-Al$_2$O$_3$-H$_2$, (**a,b**) TEM, (**c**) TEM-EDS elemental mapping, (**d**) HAADF-STEM image, (**e**) the number of atoms for the nearest neighbour distance from 532 single atoms (representative images were shown in Supplementary Fig. 8). For 0.2Pt/m-Al$_2$O$_3$-O$_2$, (**f**) an HAADF-STEM image, (**g**) the circle and square regions with the same colour and shape from **f**, (**h**) line-scanning intensity profiles obtained on the two zoomed areas in **g**. Scale bar, 200 nm (**a**), 20 nm (**b**), 100 nm (**c**), 2 nm (**d**), 2 nm (**f**).

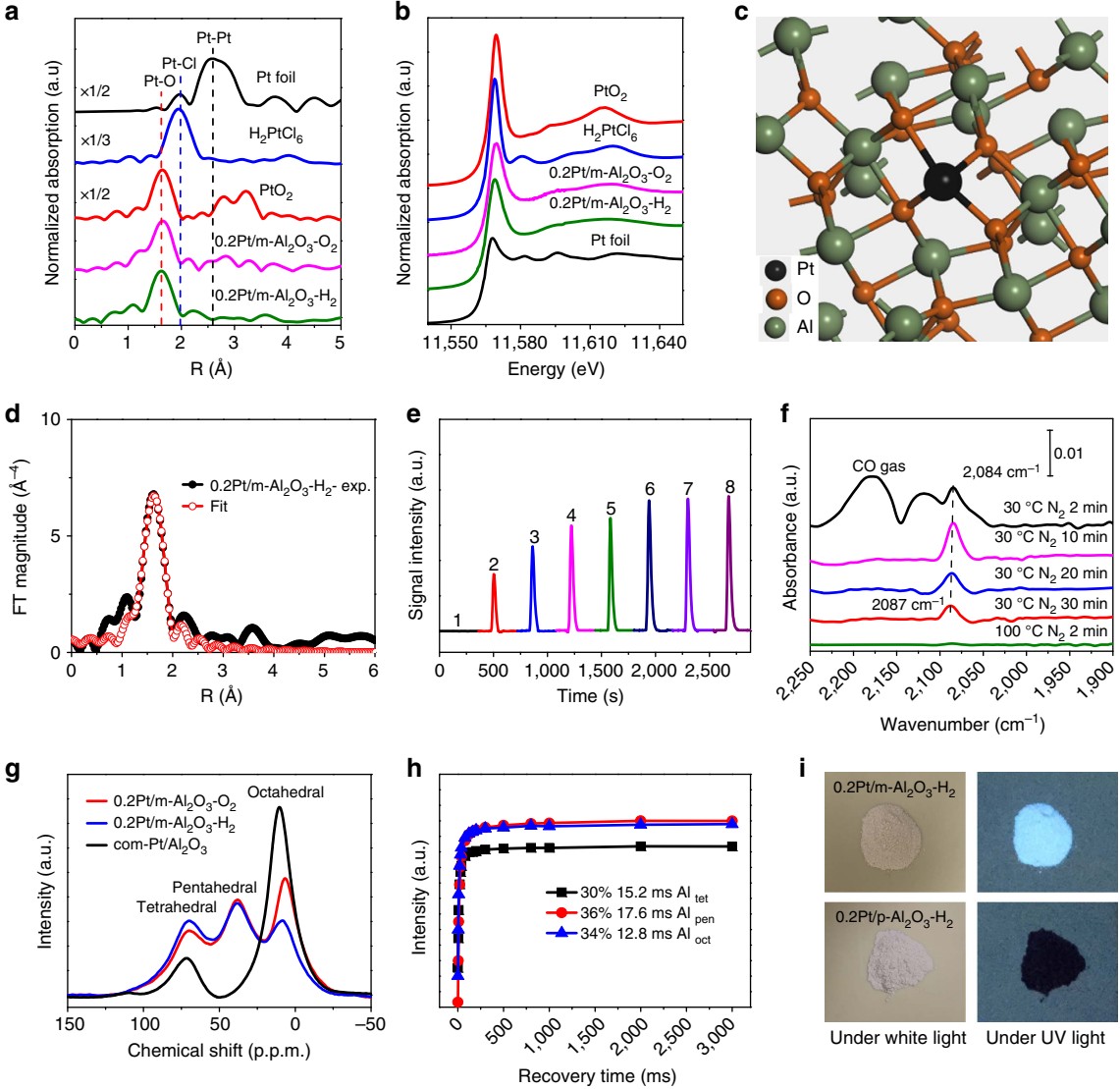

**Figure 3 | Characterization of the Pt single-atom and other control samples.** (**a**) The $k^3$-weighted Fourier transform of EXAFS spectra derived from EXAFS, $\triangle = 3.0–12.0\,\text{Å}^{-1}$, (**b**) normalized XANES spectra at the Pt L$_3$-edge of Pt foil, $H_2PtCl_6$, and $PtO_2$, 0.2Pt/m-Al$_2$O$_3$-O$_2$ and 0.2Pt/m-Al$_2$O$_3$-H$_2$, (**c**) schematic illustration of individual Pt atom located on the surface of m-Al$_2$O$_3$ for sample 0.2Pt/m-Al$_2$O$_3$-H$_2$, (**d**) FT-EXAFS curves between the experimental data and the fit, (**e**) H$_2$-O$_2$ titration profiles, (**f**) IR spectra of CO adsorbed after the desorption processes for 0.2Pt/m-Al$_2$O$_3$-H$_2$, (**g**) the $^{27}$Al MAS-NMR spectra of 0.2Pt/m-Al$_2$O$_3$-O$_2$, 0.2Pt/m-Al$_2$O$_3$-H$_2$, and commercial Pt/Al$_2$O$_3$, (**h**) relative intensity changes of tetra-, penta- and octa-coordinated Al$_2$O$_3$ with recovery time for a spin-lattice relaxation measurement of Al$_2$O$_3$ for 0.2Pt/m-Al$_2$O$_3$-H$_2$, (**i**) photographs of 0.2Pt/m-Al$_2$O$_3$-H$_2$ and 0.2Pt/p-Al$_2$O$_3$-H$_2$ under visible light and UV light (365 nm).

Extended X-ray absorption fine structure spectroscopy (EXAFS) provides important evidence on the dispersion of Pt species on m-Al$_2$O$_3$. The Pt–Pt contribution at about 2.7 Å is absent in the $k^3$-weighted EXAFS at the Pt L$_3$-edge for either 0.2Pt/m-Al$_2$O$_3$-H$_2$ or 0.2Pt/m-Al$_2$O$_3$-O$_2$ (Fig. 3a), strongly indicating that Pt exists predominantly as isolated atoms. The only prominent shell located at approximately 1.7 Å arises from Pt-O contribution, which is consistent with HAADF-STEM observation. At increased Pt loading (0.5 and 2.0 wt%), both Pt-O and Pt-Pt shells are present (Supplementary Fig. 12). The FT-EXAFS curves and the fitting results are summarized in Fig. 3d, Supplementary Fig. 13, Supplementary Table 4. The fitted Pt–O coordination number for both 0.2Pt/m-Al$_2$O$_3$-O$_2$ and 0.2Pt/m-Al$_2$O$_3$-H$_2$ is 3.8–4.0, implying that each Pt atom coordinates with four oxygen atoms on the support regardless of whether the sample has been reduced or not[36]. Notably, the four-coordination mode is inherently preferred by zero or low

valent Pt complexes, which partially explains the high stability of the Pt single-atom catalyst in the system. The electronic state of Pt species was investigated by X-ray absorption near edge structure analysis (Fig. 3b). The spectra exhibit a decreasing trend in the white-line intensities at 11,568 eV following this order: PtO$_2$ (2.20) > H$_2$PtCl$_6$ (2.08) > 0.2Pt/m-Al$_2$O$_3$-O$_2$ (1.66) ≈ 0.2Pt/m-Al$_2$O$_3$-H$_2$ (1.65) > Pt foil (1.25) (Supplementary Fig. 14)[37,38]. This observation confirms that Pt species in these catalysts are positively charged by charge transfer from Pt to Al$_2$O$_3$, in excellent agreement with EXAFS fitting results where one Pt atom is anchored on the surface of m-Al$_2$O$_3$ by coordinating with four oxygen, as illustrated in Fig. 3c.

H$_2$–O$_2$ titration was applied to probe the available Pt atoms on the surface, involving treatment of the freshly prepared sample with air, followed by titration with H$_2$. Pure m-Al$_2$O$_3$ could not absorb H$_2$ (Supplementary Fig. 15a), suggesting that any H$_2$ uptake is due to the presence of Pt. During titration, 0.2Pt/

m-$Al_2O_3$-$O_2$ and 0.2Pt/m-$Al_2O_3$-$H_2$ (Fig. 3e, Supplementary Fig. 15b) consumed 2.7 equivalent $H_2$ per mole of Pt, much higher than the samples with 0.5 and 2.0 wt% Pt loading (0.8–1.3 equivalent $H_2$ per mole of Pt) (Supplementary Fig. 15c–f), demonstrating the excellent dispersion and reactivity of Pt atoms towards $O_2$ and $H_2$ at 0.2 wt% Pt loading. The CO adsorption behaviour of the samples using DRIFTS was also investigated. For 0.2Pt/m-$Al_2O_3$-$O_2$ and 0.2Pt/m-$Al_2O_3$-$H_2$ (Fig. 3f, Supplementary Fig. 16a), a narrow, quasi-symmetrical band at around 2,087 $cm^{-1}$ is observed, which can be rationally ascribed as linearly adsorbed CO on Pt[39,40]. For Pt nanoparticles supported on α-$Al_2O_3$, CO stretching frequencies below 2,100 $cm^{-1}$ have been rigorously assigned to metallic adsorption sites on nanoparticle surfaces at either well-coordinated (∼2,090 $cm^{-1}$) or undercoordinated (∼2,070 $cm^{-1}$) Pt sites[41], but CO adsorbed on $Pt^{\delta+}$ single-atom sites may also exhibit a peak in this region. In fact, the adsorption strength and DRIFT-IR peak of CO on $Pt_1$ sites are highly dependent on the system. While $Pt_1$/ZSM-5 prepared by atomic layer deposition and $Pt_1$/$CeO_2$ prepared by atomic vapour trapping exhibited strong affinity with CO, with peak positioned at 2,095–2,115 $cm^{-1}$, $Pt_1$/$FeO_x$ synthesized via sol-gel method showed much weaker CO binding strength at 2,080 $cm^{-1}$, which is similar to our catalyst. CO readily desorbed from 0.2Pt/m-$Al_2O_3$-$H_2$ even at room temperature, with 3 $cm^{-1}$ red shift observed, which reflects the atomically dispersed Pt species may have non-identical coordination environments. For samples 0.5Pt/m-$Al_2O_3$-$O_2$, 0.5Pt/m-$Al_2O_3$-$H_2$, 2.0Pt/m-$Al_2O_3$-$O_2$ and 2.0Pt/m-$Al_2O_3$-$H_2$ (Supplementary Fig. 16b–e), two bands belonging to linearly bonded CO on $Pt^0$ nanoparticles (2,058 $cm^{-1}$) and on $Pt^{\delta+}$ single-atoms (2,087 $cm^{-1}$) are observed. The CO absorption band dropped to ca. 1/3 of its original height after 30 min purging with $N_2$ at room temperature, and disappeared after heating at 100 °C for 2 min. This suggests weak interaction between adsorbed CO and Pt species, which is consistent with the CO titration experiment where CO adsorption and desorption were found in equilibrium (Supplementary Fig. 17). All these characterizations combined provide compelling evidence that our protocol affords 0.2 wt% Pt on m-$Al_2O_3$ bearing predominant single-atom $Pt^{\delta+}$ species with exceptional high thermal stability and reactivity under both $O_2$ and $H_2$ atmospheres.

The strong interaction between Pt single-atoms and the support matrix is mainly attributed to the complex effect of $H_2PtCl_6$, aluminium isopropoxide ($C_9H_{21}AlO_3$) and P123. In an ethanol solution containing the P123 template, $C_9H_{21}AlO_3$ molecules can coordinate with $PtCl_3^-$ ions that are transformed from $PtCl_6^{2-}$ ions via ethanol reduction, forming a colourless complex (photo of these mixture solutions in Supplementary Fig. 18 and electrospray ionization mass spectrometry spectra in Supplementary Fig. 19). High dosage of the $C_9H_{21}AlO_3$ complex agent helps to restrain the Pt atoms through effectively coordinating $Al^{3+}$ species, preventing aggregation during self-assembly and calcination. Indeed, a mixture of Pt single-atoms and nanoparticles was obtained on m-$Al_2O_3$ (0.2 wt% Pt) by the conventional wet-impregnation method (sample abbreviated as 0.2Pt/m-$Al_2O_3$-imp)—the only difference being that the Pt precursor was introduced on pre-formed m-$Al_2O_3$ instead of being added at the beginning (Supplementary Figs 20 and 21).

The $Al_2O_3$ mesoporous structure and the associated unique surface environment play a vital role on the stability of atomically dispersed Pt. Three prominent peaks at 7, 38 and 70 p.p.m. are present in the $^{27}Al$ MAS NMR spectra (Fig. 3g) for 0.2Pt/m-$Al_2O_3$-$H_2$ and 0.2Pt/m-$Al_2O_3$-$O_2$, which can be assigned as $Al^{3+}$ species in tetrahedral ($AlO_4$ at 7–10 p.p.m.), pentahedral ($AlO_5$ at 38 p.p.m.) and octahedral ($AlO_6$ at 70 p.p.m.) coordination, respectively[42]. It is worth noting that only two signals at 10 and

70 p.p.m. are present for commercial Pt/$Al_2O_3$. By comparing with the $^{27}Al$ MQ NMR spectra of the gel and $C_9H_{21}AlO_3$, we find that pentahedrally coordinated $Al^{3+}$ species mainly form in calcination and reduction processes (Supplementary Fig. 22a). As such, the presence of a high concentration of pentahedrally coordinated $Al^{3+}$ species, accounting for more than one-third of $Al^{3+}$ species in m-$Al_2O_3$ based on recovery time and two-dimensional $^{27}Al$ multiple quantum NMR measurement (Fig. 3h, Supplementary Fig. 22b–d), is a direct consequence of the high quantity of defects induced by the mesoporous structure of the framework. Further NMR analysis comparing the spectra of fresh, dry sample and the sample after exposure to air revealed that a majority of $Al^V$ species stay in the bulk (Supplementary Fig. 22f,g). Despite of its low abundance on surface, $Al^V$ species may be critical to keep 0.2 wt% Pt atomically dispersed. Indeed, it has been elegantly and well established that unsaturated pentacoordinate $Al^{3+}$ centres on $Al_2O_3$ surface can strongly anchor atomically dispersed Pt species[35,42]. Considering there is no direct bonding between Al and Pt in EXAFS spectrum, Pt atoms plausibly bind to these sites via oxygen bridges[35]. We further synthesized Pt supported on disordered porous $Al_2O_3$ without adding the P123 template as a control sample (abbreviated as p-$Al_2O_3$). p-$Al_2O_3$ could not withstand high temperature treatment with $H_2$, resulting in the collapse of pores together with the agglomeration of supported Pt species (TEM images, XRD patterns, BET measurement, CO adsorbed IR spectra and $H_2$-$O_2$ titration in Supplementary Figs 23–25). Interestingly, 0.2Pt/m-$Al_2O_3$-$H_2$ is a luminescent material under UV light irradiation whereas 0.2Pt/p-$Al_2O_3$-$H_2$ is not (Fig. 3i)[32].

The thermal stability of 0.2 wt% Pt supported on m-$Al_2O_3$ and p-$Al_2O_3$ was further investigated by calcination at 600 and 800 °C in air for 4 h. XRD patterns, BET analysis and their pore size distribution curves (Supplementary Fig. 26), TEM images (Supplementary Fig. 27)[27], Al MAS-NMR spectra and IR spectra of CO absorption (Supplementary Fig. 28) indicate that m-$Al_2O_3$ preserved its mesoporous structure and surface area, with a majority of Pt species maintaining single-atom identities. Noteworthy, Pt can be emitted as volatile $PtO_2$ at 800 °C under oxidative conditions[31]. However, the Pt content remained at around 0.2 wt% after thermal treatments at 600 and 800 °C (Supplementary Table 1), highlighting Pt is strongly anchored in m-$Al_2O_3$ in the material. On the other hand, p-$Al_2O_3$ exhibited decreased surface area, diminished pentahedral unsaturated $Al^{3+}$ centres and collapsed porous structure after high temperature treatment. Meanwhile, the Pt species in p-$Al_2O_3$ was largely transformed into nanoparticles. These control experiments indicate that the mesoporous structure and the unique interior surface framework of m-$Al_2O_3$ play a decisive role for efficient stabilization of atomically dispersed Pt.

**Catalytic activity: Selective hydrogenation reactions.** The activity and selectivity of single-atom Pt catalysts were investigated over a series of reactions in the presence of $H_2$, including the hydrogenation of nitrobenzene[43], acetophenone[44], phenylacetylene[36] and 1,3-butadiene[33] (Supplementary Fig. 30 and Supplementary Tables 5–7). In all cases, excellent yields of desired products were obtained. For instance, 98.7% selectivity to 1-phenylethanol in the hydrogenation of acetophenone was achieved over 0.2Pt/m-$Al_2O_3$-$H_2$, which is higher than 0.2Pt/p-$Al_2O_3$-$H_2$ (78.9%) and commercial Pt/$Al_2O_3$ (68.3%) (Supplementary Table 5). The excellent activity and selectivity not only highlight the applicability of the catalyst for hydrogenation reactions, but are encouraging testimonies of the single-atom identity and the positively charged nature of Pt species on m-$Al_2O_3$ (ref. 36). Aromatic ring normally coordinates

to multiple metal atoms before undergoing hydrogenation, and remain to interact with metal surface during step-wise hydrogenation[45]. This mechanism is not possible over single-atom catalysts, which satisfactorily explains why ring hydrogenation on 0.2Pt/m-Al$_2$O$_3$-H$_2$ was almost fully suppressed. In contrast, positively charged Pt species favours C=O bond adsorption and activation forming an $\eta^1$(O) configuration, as previously established on Pt nanoparticle catalysts[46,47]. In our system, $\eta^1$(O) acetophenone adsorbed on Pt single-atom site plausibly reacts with spilled-over hydrogen to form 1-phenylethanol. Afterwards the product transfers to Al$_2$O$_3$ support where it is more strongly adsorbed before finally diffuses into the solution phase[47], leading to regeneration of the Pt site for a new cycle.

Next, selective hydrogenation of 1,3-butadiene were studied in detail to investigate the stability of our single-atom Pt catalysts in reductive atmosphere at elevated temperatures. 0.2Pt/m-Al$_2$O$_3$-H$_2$ transformed 1,3-butadiene into butenes with >99% selectivity at 50 °C, without affecting the co-feed propylene molecule. The TOF for butane formation is 0.034 s$^{-1}$, a few times higher than earlier reported Pt$_1$Cu catalyst under comparable condition (Supplementary Table 8)[33]. 0.2Pt/p-Al$_2$O$_3$-H$_2$ and commercial Pt/Al$_2$O$_3$ catalysts, on the other hand, exhibited much lower selectivity (77 and 29%, respectively) (Fig. 4a, Supplementary Figs 31 and 32). To evaluate the long-term stability of these catalysts under H$_2$ atmosphere at high temperature, the three catalysts were exposed to the mixture of gas reagents at 200 °C for 24 h (Fig. 4b, Supplementary Figs 33–35). Afterwards, their catalytic performances were re-evaluated at 30 °C for 12 h. Remarkably, 0.2Pt/m-Al$_2$O$_3$-H$_2$ even exhibited slightly increased activity and near-quantitative selectivity towards butenes, confirming the preservation of Pt active sites after high temperature treatment (Fig. 4c, Supplementary Figs 33–35), whereas 0.2Pt/p-Al$_2$O$_3$-H$_2$ and commercial Pt/Al$_2$O$_3$ dropped in activity and selectivity. To our delight, a number of well-separated Pt atoms are clearly identified by HAADF-STEM on spent 0.2Pt/m-Al$_2$O$_3$-H$_2$ catalyst, with no visible Pt atom aggregation being observed (Fig. 4d, Supplementary Figs 36 and 37). The IR spectrum of CO adsorption on 0.2Pt/m-Al$_2$O$_3$-H$_2$ shows a single sharp peak at 2,090 cm$^{-1}$, substantiating HAADF-STEM finding that Pt species remain predominantly isolated. The major CO absorption peak on spent 0.2Pt/p-Al$_2$O$_3$-H$_2$ catalyst, on the other hand, shifts to 2,064 cm$^{-1}$ suggesting significant formation of Pt nanoparticles (Fig. 4e).

**Catalytic activity: *n*-hexane reforming.** Most Pt catalysts for high temperature applications under reductive conditions are related to oil refining. The hydro-reforming of *n*-hexane provides a good model reaction to understand the catalytic conversion of linear hydrocarbons into branched isomers under H$_2$ which is a vital reaction for fuel and chemical production from naphtha. Typically, *n*-hexane reacts via four major pathways: cracking to shorter chain hydrocarbons; isomerization to 2-methylpentane, 3-methylpentane and multi-branched isomers; cyclization to methylcyclopentane or cyclohexane; and aromatization to benzene (Supplementary Fig. 38)[48]. One critical requirement in this process is to achieve long durability against deactivation via preventing carbon deposition. Previous studies were focused on Pt nanoparticles[49], whereas little attention has yet been paid to single-atom catalysts. We compared the performance of 0.2Pt/m-Al$_2$O$_3$-H$_2$, 0.2Pt/p-Al$_2$O$_3$-H$_2$, and commercial Pt/Al$_2$O$_3$ catalysts for *n*-hexane reforming at 400 and 550 °C. The conversion of *n*-hexane was maintained below 30% in all cases to ensure that the reaction is under kinetic control. While the initial activities (reflected by *n*-hexane conversion) of the three

catalysts at both 400 and 550 °C were similar, the selectivity of 0.2Pt/m-Al$_2$O$_3$-H$_2$ towards desired isomeric products was the highest among the three (ca. 50%), in particular at 550 °C (Fig. 4f, Supplementary Figs 39 and 40). The superior selectivity of 0.2Pt/m-Al$_2$O$_3$-H$_2$ catalyst for branched products is likely due to the strong interaction between Pt and support, which is known to be critical for *n*-hexane isomerization[29].

0.2Pt/m-Al$_2$O$_3$-H$_2$, 0.2Pt/p-Al$_2$O$_3$-H$_2$ and commercial Pt/Al$_2$O$_3$ catalysts were subjected to long-term stability tests. Under both 400 and 550 °C, the stability of the 0.2Pt/m-Al$_2$O$_3$-H$_2$ catalyst, reflected by the substrate conversion, was superior than other catalysts. Its final activity merely dropped to 88% compared to the activity of the fresh catalyst after reacting at 550 °C for 48 h. In contrast, a staggering decrease of activity to 50% for 0.2Pt/p-Al$_2$O$_3$-H$_2$ after 48 h and to 38% for commercial Pt/Al$_2$O$_3$ after 24 h was observed (Fig. 4f). The deactivation rates of the three catalysts were generated by curve fitting the conversion as a function of time, followed by differentiation. 0.2Pt/m-Al$_2$O$_3$-H$_2$ exhibited a horizontal line suggesting negligible deactivation whereas 0.2Pt/p-Al$_2$O$_3$-H$_2$ and commercial Pt/Al$_2$O$_3$ catalysts continued to drop in activity (Fig. 4g). The selectivity for isomeric products remained relatively constant for all samples, with 0.2Pt/m-Al$_2$O$_3$-H$_2$ being the most selective (Fig. 4h,i).

There are no detectable Pt nanoparticles in the TEM images for 0.2Pt/m-Al$_2$O$_3$-H$_2$ after reacting for 48 h at 400 and 550 °C (Supplementary Fig. 41), but a number of Pt nanoparticles were identified on 0.2Pt/p-Al$_2$O$_3$-H$_2$ (Supplementary Fig. 42) and commercial Pt/Al$_2$O$_3$ (Supplementary Figs 29 and 43) after the reaction. In addition, 0.2Pt/p-Al$_2$O$_3$-H$_2$ had more carbon deposition than 0.2Pt/m-Al$_2$O$_3$-H$_2$ from TEM images, thermo-gravimetry (TG) measurements (Supplementary Fig. 44) and visual inspection (Supplementary Fig. 45). BET analysis and $^{27}$Al MAS-NMR spectroscopy reveal insignificant changes of meso-pores as well as Al species on 0.2Pt/m-Al$_2$O$_3$-H$_2$. Serious damage of the porous structure on 0.2Pt/p-Al$_2$O$_3$-H$_2$ was however observed after reaction (BET analysis in Supplementary Fig. 46, $^{27}$Al MAS-NMR spectra in Supplementary Fig. 47). Nevertheless, obvious changes in CO adsorption IR spectra are found for all samples. The CO absorption band broadened on 0.2Pt/m-Al$_2$O$_3$-H$_2$, with the main peak shifting from 2,089 to 2,084 cm$^{-1}$, together with a shoulder peak at 2,060 cm$^{-1}$ (Supplementary Fig. 48). Meanwhile, there is a significant drop of Pt–O coordination number from 3.6 to 1.1 (after reaction at 400 °C) and 1.4 (after reaction at 550 °C), with concurrent increase of Pt–Pt coordination number to around 6 (Supplementary Fig. 49 and Supplementary Table 4). These suggested the formation of Pt$^0$ nanoparticles/nanoclusters after reaction, but there is still a substantial amount of isolated Pt atoms in the catalyst. On the other hand, only a single band at 2,060 cm$^{-1}$ can be observed for commercial Pt/Al$_2$O$_3$ and 0.2Pt/p-Al$_2$O$_3$-H$_2$ catalysts, ascribed to linearly bonded CO on Pt$^0$ sites on nanoparticles (Supplementary Fig. 50).

**Catalytic activity: CO oxidation.** *CO oxidation.* To evaluate the catalyst stability under oxidative conditions, CO oxidation between 100 and 400 °C was conducted. While thermally stable Pt nanoparticle catalysts for CO oxidation have been developed[30], high temperature Pt single-atom catalysts have been much less studied[31]. A feed gas containing 2.5 vol% CO, 2.5 vol% O$_2$ and balance Ar was passed through the reactor at a flow rate of 80 ml min$^{-1}$ (corresponding to a space velocity of 4.8 × 10$^7$ ml h$^{-1}$ g$^{-1}_{pt}$, Supplementary Figs 51 and 52). The TOF was 0.023 s$^{-1}$ at 200 °C and steadily increased to 0.175 s$^{-1}$ at 250 °C (Supplementary Table 9). These numbers well match

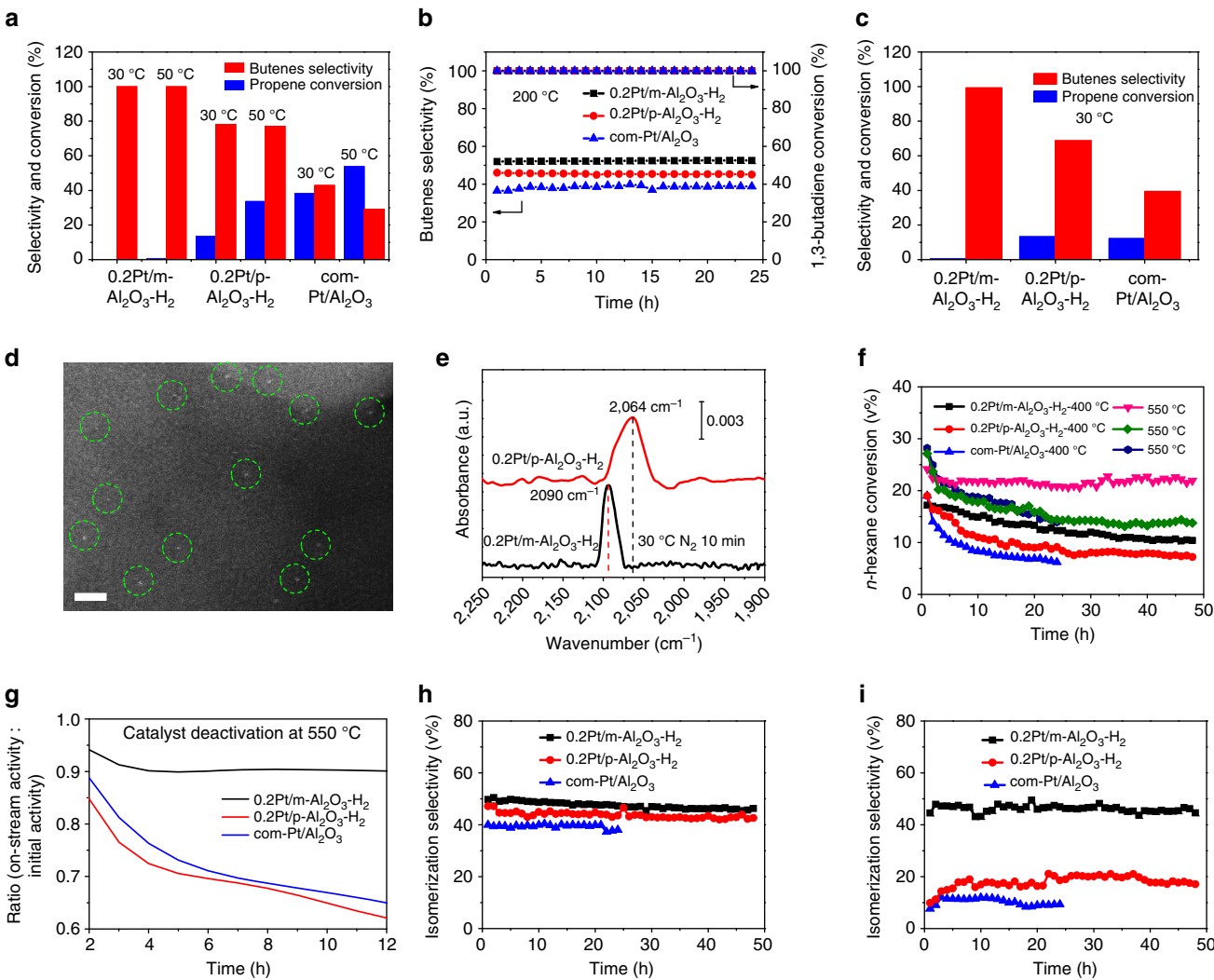

**Figure 4 | Evaluation of 0.2Pt/m-Al₂O₃-H₂ and control catalysts in 1,3-butadiene hydrogenation and n-hexane reforming reaction.** (**a**) The selectivity of butenes and conversion of propene at 30 and 50 °C, (**b**) the selectivity of butenes and the conversion of 1,3-butadiene at 200 °C for 24 h, (**c**) the selectivity of butenes and conversion of propene at 30 °C for 0.2Pt/m-Al₂O₃-H₂, 0.2Pt/p-Al₂O₃-H₂ and commercial Pt/Al₂O₃, after the entire reaction sequence, (**d**) an HAADF-STEM image for 0.2Pt/p-Al₂O₃-H₂ after the entire reaction sequence, and (**e**) CO adsorption IR spectra for 0.2Pt/m-Al₂O₃-H₂, 0.2Pt/p-Al₂O₃-H₂ after the entire reaction sequence, (**f**) the conversion of n-hexane at 400 and 550 °C, (**g**) the catalysts deactivation at 550 °C, (**h**) the isomerization selectivity at 400 °C, and (**i**) the isomerization selectivity at 550 °C for 0.2Pt/m-Al₂O₃-H₂, 0.2Pt/p-Al₂O₃-H₂, commercial Pt/Al₂O₃ catalysts. The reaction was conducted in a gas mixture of 2% 1,3-butadiene, 20% propene, 16% H₂ and balance He at a flow rate of 20 ml min⁻¹, 0.1 ml h⁻¹ n-hexane and 6 ml min⁻¹ pure H₂ at ambient pressure. Scale bar, 2 nm (**d**).

those of a Pt₁ catalyst on θ-Al₂O₃ (0.013 s⁻¹ at 200 °C and 0.187 s⁻¹ at 250 °C)[50]. In the recycling experiment (100–400 °C), the activity of 0.2Pt/m-Al₂O₃-H₂ increased slightly in the first 13 rounds, and then became very stable in the subsequent 37 cycles (Fig. 5a, Supplementary Fig. 51). The temperature of the reactor was then fixed at 400 °C for 220 h, during which 100% CO conversion was maintained for the entire period (Fig. 5b). Afterwards, the 0.2Pt/m-Al₂O₃-H₂ spent catalyst was inspected again in recycling experiments (100–400 °C). The conversion-temperature curves for 50th–60th cycles fully resemble those collected before long-term treatment at 400 °C, and this convincingly demonstrates exceptionally high catalytic stability (Fig. 5c). Finally, the temperature of the reactor was set at 230 °C—deliberately selected for an incomplete CO conversion—and maintained for 70 h (Fig. 5d). No appreciable drop in CO conversion was detected.

The spent 0.2Pt/m-Al₂O₃-H₂ catalyst after the above-mentioned reaction sequence—corresponding to an exposure to reaction conditions for over a month—was thoroughly interrogated

by a series of instrumental analysis. No Pt nanoparticles can be detected from the TEM image and TEM-EDS elemental mapping, indicating that Pt species is uniformly dispersed on spent 0.2Pt/m-Al₂O₃-H₂ (Supplementary Fig. 53a–d). The CO absorption peak remains sharp and quasi-symmetrical, centred at 2,090 cm⁻¹ (Fig. 5e). In the EXAFS spectrum, Pt–O contribution located at approximately 1.7 Å remains as the only prominent shell, unarguably proving that Pt largely maintains single-atom identity (Fig. 5f, Supplementary Fig. 53e). While the white line intensity (1.64) and Pt–O coordination number (3.6) remain almost identical after reaction, an increase in Debye–Waller factor was observed, reflecting some degree of evolution of catalyst active centre (Supplementary Table 4). BET analysis and the ²⁷Al MAS-NMR spectrum confirm the integrity of the mesoporous structure, and the preservation of a majority of unsaturated pentahedral Al³⁺ species (Supplementary Figs 53f and 54).

Similar recycling results were observed over 0.2Pt/m-Al₂O₃-H₂ catalyst under dilute conditions (100 mg 0.2Pt/m-Al₂O₃-H₂ mixed with 1.0 g commercial Al₂O₃, Supplementary Fig. 55).

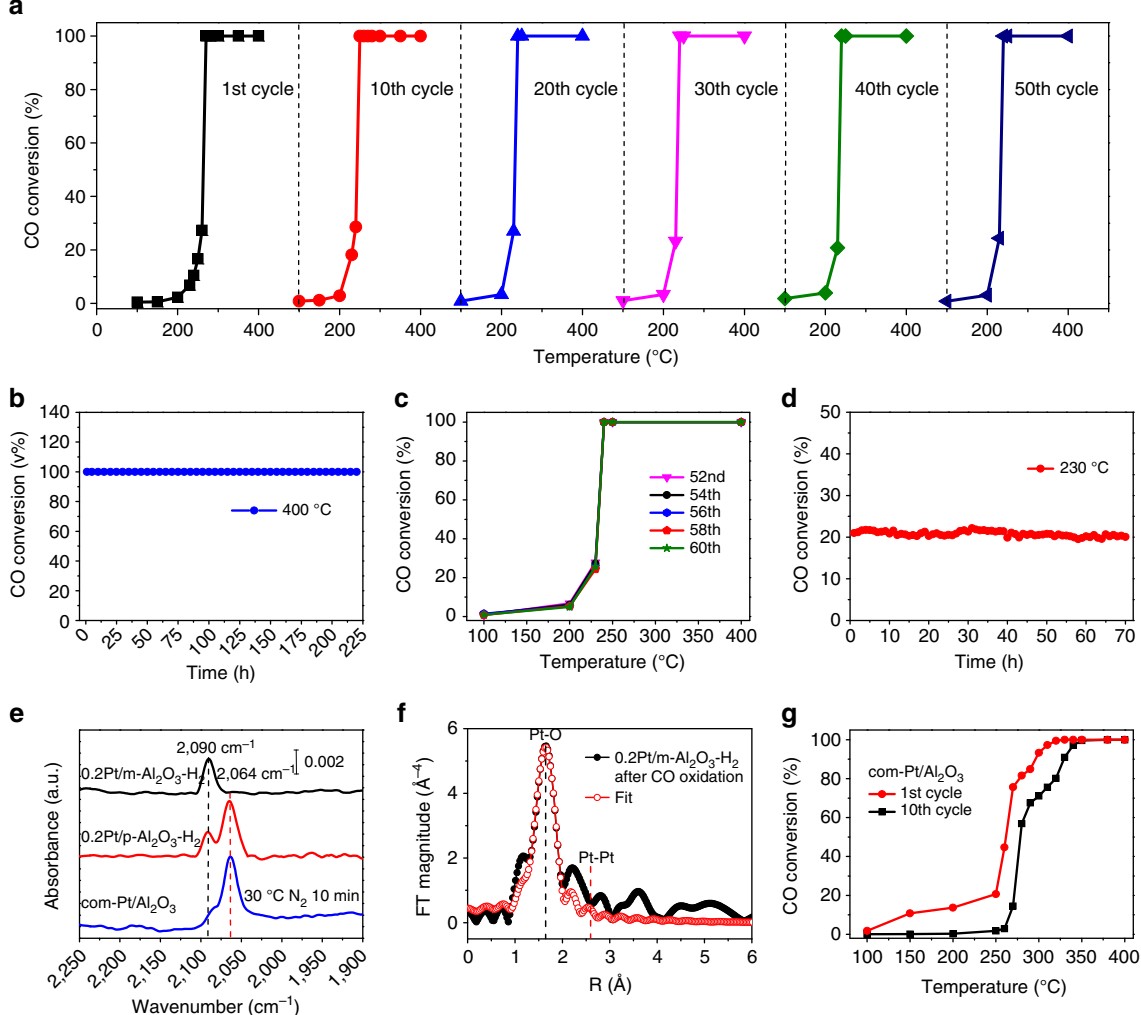

**Figure 5 | Evaluation of 0.2Pt/m-Al$_2$O$_3$-H$_2$ and control catalysts in CO oxidation.** (**a**) Conversion of CO from 100 to 400 °C with 1st–50th cycles, (**b**) maintained at 400 °C for 220 h, (**c**) another ten cycles from 100 to 400 °C, (**d**) maintained at 230 °C for 70 h after 60 cycles, (**e**) IR spectra for 0.2Pt/m-Al$_2$O$_3$-H$_2$ after the CO oxidation sequence as shown in **a–d**, 0.2Pt/p-Al$_2$O$_3$-H$_2$ after 14 cycles CO oxidation and commercial Pt/Al$_2$O$_3$ catalysts after ten cycles CO oxidation, (**f**) the $k^3$-weighted Fourier transform spectrum derived from EXAFS for 0.2Pt/m-Al$_2$O$_3$-H$_2$ after the CO oxidation sequence as shown in **a–d**, (**g**) conversion of CO from 100 to 400 °C in the 1st and the 10th cycles over commercial Pt/Al$_2$O$_3$ (named com-Pt/Al$_2$O$_3$). 2.5 vol% CO, 2.5 vol% O$_2$ and balance Ar was allowed to pass through the reactor at a flow rate of 80 ml min$^{-1}$.

0.2Pt/m-Al$_2$O$_3$-O$_2$, too, was highly stable in CO oxidation despite a slightly lower activity (Supplementary Fig. 56). Moreover, after calcined at 600 °C the catalyst remained high stability over 14 cycles (0.2Pt/m-Al$_2$O$_3$-O$_2$-600) (Supplementary Fig. 28e). In comparison, the catalytic activity of commercial Pt/Al$_2$O$_3$, 0.2Pt/p-Al$_2$O$_3$-H$_2$ and 0.2Pt/m-Al$_2$O$_3$-imp decreased only after several cycles (100–400 °C) (Fig. 5g, Supplementary Figs 57 and 58). The Pt species underwent agglomeration in these control samples to form larger particles as shown in TEM images (Supplementary Fig. 59). BET and TEM analysis also suggest decreased surface area with collapsed porous structures. The CO adsorption experiment on spent commercial Pt/Al$_2$O$_3$ and 0.2Pt/p-Al$_2$O$_3$-H$_2$ reveals the relative decrease of the peak at 2,090 cm$^{-1}$ with increase of the peak at about 2,064 cm$^{-1}$, corroborating TEM findings (Fig. 5e).

## Discussion

The metal–support interactions often play a pivotal role in shaping the stability and reactivity of supported metal catalysts[10,11,51,52]. This effect is more pronounced and even becomes the determining factor in single-atom catalysts,

considering isolated metal atoms are in sole contact with the support. Strong interactions between the metal atom and the support are conceived to be the key to preventing metal atom aggregation[6], while an overly strong interaction may lead to catalytically inactive species acting as spectators. In some earlier works[23,36,43], atomically dispersed Pt catalysts were obtained at low-temperature reduction conditions, and could be used in mild hydrogenation reactions. Unfortunately, atom aggregation into clusters and nanoparticles could occur when the catalysts are exposed to increased reduction temperature, or employed in harsher reaction conditions, plausibly due to insufficient anchoring effect of the support to Pt atoms.

In the current work, we have developed an atomically dispersed Pt catalyst supported on mesoporous Al$_2$O$_3$ exhibiting astounding catalytic activity and stability under both oxidative and reductive atmospheres at high temperatures. The complex effect between the metal and support precursors enables the Pt cation to be strongly anchored in the inner surface of the support, staying in a four oxygen coordination mode that is intrinsically favoured by low valent Pt atoms. Additionally, the P123 template facilitates the formation of highly stable mesoporous structures

enriched with unsaturated pentahedral $Al^{3+}$ centres, further enhancing metal–support interactions. As a result, the single-atom catalyst can survive for long durations in a series of reactions under demanding conditions. An apparent limitation of the system is that the single-atom identity of Pt was achieved only at a low loading (0.2 wt%), plausibly due to the low abundance of $Al^V$ species on surface. Nevertheless, our work adds solid proof that the single-atom catalyst may not be necessarily more vulnerable than nanocluster and nanoparticle catalysts under harsh reaction conditions. Rational control of the structure and surface property of the support, and fine-tuning of the synthetic procedure, are two key factors to achieve desirable metal atom–support interactions in atomically dispersed Pt catalysts for high temperature operations with superior activity and stability. For the current system, future work should be directed to enriching $Al^V$ species on the surface by tuning the synthetic procedure, and/or incorporate other metal oxides such as silica and titania to the alumina support.

## Methods

**Catalysts synthesis.** In a typical synthesis, Pluronic P123 (2.1 g) was dissolved in ethanol (20 ml) at room temperature. Then, 67% nitric acid (3.2 ml) and aluminium isopropoxide (4.08 g) were also dissolved in ethanol (20 ml). The two solutions were mixed under vigorous stirring[34]. Next, stoichiometric amounts of $H_2PtCl_6$ (0.2, 0.5, and 2.0 wt% of Pt respectively, compared with $Al_2O_3$ to be generated) dissolved in ethanol (0.0193 mol l$^{-1}$) were added to the mixture under stirring. The solution was covered with a PE film and stirred at room temperature for 48 h and then placed in an oven at 60 °C for ethanol evaporation for 72 h. The final gel was calcined at 400 °C for 4 h at a heating rate of 1 °C min$^{-1}$, named as 0.2Pt/m-$Al_2O_3$-$O_2$, 0.5Pt/m-$Al_2O_3$-$O_2$ and 2.0Pt/ m-$Al_2O_3$-$O_2$. The samples (0.2Pt/m-$Al_2O_3$-$O_2$, 0.5Pt/m-$Al_2O_3$-$O_2$ and 2.0Pt/m-$Al_2O_3$-$O_2$) were reduced in 5% $H_2$/$N_2$ at 400 °C for 1 h at a heating rate of 5 °C min$^{-1}$. The resulting samples are named as 0.2Pt/m-$Al_2O_3$-$H_2$, 0.5Pt/m-$Al_2O_3$-$H_2$ and 2.0Pt/m-$Al_2O_3$-$H_2$, respectively.

**Catalysts characterization.** Wide-angle XRD analysis was carried out using a Bruker D8 Advance X-Ray Diffractometer, at a scan rate of 2° min$^{-1}$. It was operated at 40 kV applying a potential current of 30 mA. The microscopic features of the samples were characterized using a TEM equipped with EDX (JEM-2100F) operated at 200 kV. STEM images were acquired on a Cs-corrected FEI Titan G2 60–300 Microscope operated at 300 kV using an HAADF detector. TG analysis was performed using an EXSTAR TG/DTA 6300 (Seiko Instruments) at a heating rate of 5 °C min$^{-1}$ in air. The porous nature of the samples was investigated by measuring the physical adsorption of nitrogen at a liquid-nitrogen temperature of −196 °C using an automatic volumetric sorption analyzer (NOVA3200e, Quantachrome). Electrospray ionization mass spectrometry spectra were obtained from a Bruker MicroTOF-Q system. The samples were directly injected into the chamber at 20 μl min$^{-1}$. Typical instrument parameters: capillary voltage, 4 kV; nebulizer, 0.4 bars; dry gas, 2 l min$^{-1}$ at 120 °C; $m/z$ range, 40–3,000. The $^{27}Al$ MAS NMR measurements were carried out on a Varian 300 MHz NMR spectrometer using a 7.5 mm HX MAS probe with a spinning rate of 13 kHz at resonance frequencies of 156 MHz. ICP-OES analysis was conducted on iCAP 6000 series inductively coupled plasma optical emission spectrometry. The catalysts were digested in aqua regia (HCl/$HNO_3$ = 3:1) at 373 K for 6 h and then diluted with deionized water to a certain volume before analysis.

DRIFTS measurements were performed on a Thermo Scientific Nicolet iS50 instrument with an Hg-Cd-Te (MCT) detector, and a Praying Mantis high temperature reaction chamber with ZnSe windows. TPR, $H_2$-$O_2$ and CO titration analyses were conducted on a ChemBET Pulsar TPR/TPD (Quantachrome). Pt $L_3$-edge XAS spectra of Pt catalysts and reference samples were recorded at the BL5S2 beamline at the AichiSR (Aichi Synchrotron Radiation Center, Aichi, Japan) or at the BL01B1 beamline at the SPring-8. Data analysis was carried out with Athena and Artemis included in the Demeter package. For curve fitting analysis of EXAFS spectra, each theoretical scattering path was generated with FEFF 6.0 L. The $k^3$-weighted EXAFS oscillation in the range of 3.0–12 Å$^{-1}$ was Fourier transformed. GC analysis was performed using an Agilent 7890 GC equipped with FID detector with two capillary columns: JW-CYCLODEX-B 30 m, 0.25 mm and JW-GS-ALUMINA/KCl 50 m, 0.53 mm. Another Agilent 7890 GC is equipped with a TCD detector and also two columns: G3591-80004 6Ft HayeSep Q 80/100 and G3591-80022 8Ft MolSieve 5A 60/80. GC-MS data were obtained using an Agilent Technologies 7890 GC with 5975 MSD.

**Catalytic activity evaluation.** All reactions were evaluated in a fixed-bed reactor under ambient pressure with online GC detection, and the samples were grinded before the tests.

**CO oxidation.** 100 mg catalyst was loaded into the reactor tube. Then it was either calcined in air at 400 °C for 1 h or reduced *in-situ* with 5 vol% $H_2$/$N_2$ at 400 °C for 1 h. After cooling to room temperature, the feed gas containing 2.5 vol% CO, 2.5 vol% $O_2$ and balance Ar was allowed to pass through the reactor at a flow rate of 80 ml min$^{-1}$ (corresponding to a space velocity of $2.4 \times 10^7$ ml h$^{-1}$ g$^{-1}_{pt}$ ($6.7 \times 10^3$ ml s$^{-1}$ g$^{-1}_{pt}$), specific rate of $7.4 \times 10^{-3}$ mol$_{CO}$ g$^{-1}_{Pt}$ s$^{-1}$ (1.4 mol$_{CO}$ mol$^{-1}_{Pt}$ s$^{-1}$) or $1.5 \times 10^{-5}$ mol$_{CO}$ g$^{-1}_{cat}$ s$^{-1}$ ($2.9 \times 10^{-3}$ mol$_{CO}$ g$^{-1}_{Pt}$ s$^{-1}$)). In every reaction cycle, the temperature increased from 100 to 400 °C with ca. 50 °C interval and stabilized at each temperature for 20 min before measuring the CO conversion and the $CO_2$ generation. Once a cycle was finished, the reactor was cooled to room temperature to start a new cycle.

**Selective hydrogenation.** The as-synthesized pre-catalyst was reduced in $H_2$ at 400 °C for 1 h before the reaction. A gas mixture of 2% 1,3-butadiene, 20% propylene, 16% $H_2$ and balance He at a flow rate of 20 ml min$^{-1}$ was introduced at ascending temperature (30–200 °C). GC analysis was done at a few selected temperatures after stabilization at that point for at least 10 min. The deactivation test was performed at 200 °C for 24 h using 200 mg catalyst, after which the catalyst was re-evaluated at 30 °C.

**Reforming.** Three hundred milligrams catalysts were loaded into the reactor. The catalyst was pretreated at 400 and 550 °C for 1 h in pure $H_2$. The reforming reaction was conducted with 0.1 ml h$^{-1}$ of *n*-hexane feeding carried by a syringe pump, and 6 ml min$^{-1}$ pure $H_2$ at 400 °C or 550 °C. Each temperature was maintained for 48 h. In addition to online GC analysis, the products were also collected in an ice-cooled cold trap for GC-MS analysis.

**Data availability.** The data that support the findings of this study are available from the authors on reasonable request.

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

## Acknowledgements

We thank the National University of Singapore Young Investigator Award (WBS: R-279-000-464-133) for financial support. HAADF-STEM measurement at King Abdullah University of Science and Technology, Saudi Arabia; $^{27}$Al MAS NMR measurements were carried out at Dalian Institute of Chemical Physics, Chinese Academy of Sciences, China; the XAS measurements were performed at the BL5S2 at the Aichi Synchrotron Radiation Center under the approval of Aichi Science & Technology Foundation, Aichi, Japan (Proposal No. 201506061) and at the BL01B1 at the SPring-8 (Japan Synchrotron Radiation Research Institute, Hyogo, Japan) under the approval of JASRI (Proposal No. 2016A1025). The XAS measurement at the AichiSR was financially supported by the Nagoya University Synchrotron Radiation Research Center.

## Author contributions

N.Y. and Z.Z. conceived the idea, designed the experiments, analysed the data and wrote the manuscript. Z.Z. carried out the catalyst synthesis, evaluated their catalytic performances and conducted some characterizations. Y.H. and Y.Z carried out HAADF-STEM characterizations. T.Z., A.W. and M.Z. arranged and conducted $^{27}$Al MAS NMR characterizations and DFT simulation. T.T. and H.A. carried out the XAS measurements. B.Z. and J.Z. participated in material characterizations and catalytic reaction evaluation. All authors discussed the results and edited the manuscript.

## Additional information

**Competing interests:** The authors declare no competing financial interests.

**Publisher's note**: 

