## [Peer Review File · Nature Communications]

Reviewers' comments:

Reviewer #1 (Remarks to the Author):

This is a very thorough paper describing the synthesis, characterization, reactivity and stability of high dispersion Pt catalysts on Al₂O₃. The developed synthetic protocol introduces high concentrations of Al³⁺ sites that act as strong binding sites for atomically dispersed Pt. It is shown that the samples containing high concentrations of Al³⁺ are able to maintain good stability for the atomically dispersed Pt under a wide range of conditions. The analysis and conclusions are for the most part convincing and publication in Nat Comm is recommended after a few issues are addressed.

1. It is argued that the Al³⁺ centers are critical binding sites for the Pt atoms. However, no Pt-Al interactions were observed or discussed on the XAS data. Some discussion on this should be added.

2. The IR interpretation should be further substantiated.

a. For Pt/Al₂O₃, CO stretching frequencies below 2100 cm⁻¹ have been rigorously assigned to metallic adsorption sites on nanoparticle surfaces at either well coordinated (~2090 cm⁻¹) or undercoordinated (~2070 cm⁻¹) Pt sites (see ACS Catal. 2016, 6, 5599–5609). It is surprising that CO adsorbed to the cationic Pt single atoms don't exhibit stretching frequencies > 2100 cm⁻¹. It is known that both Pt structure and oxidation state play a role in defining the CO stretching frequency, so this should be discussed and somehow rationalized. See J. OF CATALYSIS 169, 382–388 (1997) for a nice discussion on some of these issues.

b. The CO band observed in Figures 2F and 4E are missing some signatures of adsorption at single Pt sites. First, the band positions seem to shift with coverage, suggesting dipole interactions that shouldn't exist at single atom adsorption sites. Second, the bands are quite wide in FWHM and have some obvious asymmetry that suggests there are multiple adsorption sites being observed that have different chemical characteristics.

c. In the recent Science papers by Stair's group and Datye's groups CO was observed to stick very strongly for stretches that were assigned as single Pt atom adsorption sites. This does not agree with the results presented here. This should be discussed.

3. It is mentioned that the reactivity of the 0.2 Pt-m/Al₂O₃ actually increases with time on stream. This is suggesting there may be some agglomeration of Pt atoms to form small metallic clusters that are more active for this reaction. Some suggestions of what structural or chemical changes may be occurring to the catalyst that induce this reactivity change should be added.

Reviewer #2 (Remarks to the Author):

This manuscript reports the synthesis, characterization and catalyst stability of Pt single-atom catalyst supported on mesoporous alumina. The manuscript is interesting in that the authors compare the performance of different alumina samples, exploring the role of alumina coordination on the nature of Pt species and their stability. This makes the work novel, since their synthesis involves commonly used reagents and alumina is among the most widely used catalyst supports, so the work could have broad applicability. However, the work does not represent any fundamental advance, since even transitional alumina was shown (ref 34) to stabilize single atoms. What the authors have demonstrated is that their catalyst is stable when the Pt concentration is low (0.2 wt%) preventing the Pt species from forming larger particles. At higher concentrations, the Pt tends to form larger particles. And it appears that the reactivity of these catalysts is not superior to that of the conventional catalysts. The work is not suitable for publication in its present form and needs major revisions to correct the errors in the interpretation of the data, as listed below. And the reactivity comparisons for CO oxidation need TOF data and for the selective hydrogenation, need to be compared at similar conversion.

Specific comments

- 1) The authors have synthesized mesoporous alumina (m-alumina), and then another one they call porous alumina (p-alumina). Since alumina tends to crystallize when heated, I would like to see evidence that the mesoporous alumina structure is preserved after their high temperature treatments. Supplementary figure 7 shows ordered structure in the as-prepared state. Figure S8 and S9 shows them after treatments, but the temperature is not stated, and the samples do not exhibit any order. Hence, the authors need to state whether or not the pore structure is retained. Otherwise, I would infer that the m- and p- aluminas are not probably very different after being heat treated.
- 2) The N₂ adsorption isotherms in Fig. S5 do not suggest a broad pore size distribution, not what one would expect based on the TEM images (which of course sample very small regions of the specimen). Hence, the authors need to show the low angle XRD region which will show clearly the extent of ordered mesoporosity in their structure. This is important since the authors claim that their samples are stable under extreme conditions and the m-alumina is better, so establishing the stability of the alumina pore structure is important.
- 3) The manuscript should omit reporting data from the literature, for example figure S1 on utilization of Pt and Figure 1 in the manuscript, which is not reporting any original data from this study.
- 4) The interpretation and analysis of the images in Fig. 2 are incorrect. First, to image single atoms of Pt you need a probe diameter of sub-Angstrom size. Each bright dot will then correspond to the size of the probe. I cannot tell from the scale of the image if each of those dots meets this criterion. I do not understand the Gaussian fit of image g, which should show a single peak corresponding to the size of the probe (since the atom is much smaller in size). I cannot read the scale very clearly in this rather fuzzy image, but establishing this is important to convince the reader that they truly have isolated single atoms.
- 5) Secondly, Fig. 2e is incorrect in asserting this to be the interparticle distance, since the STEM image in Figure 2 is a projection of the 3-D sample. Hence what you measure as the interatom distance is not really the true distance, it is a projection of this distance on to a plane. It is clear that the bright dots don't have the same contrast, so they are not in one plane. And to make any inferences from the interatomic distances in the cluster seen in figure 2g is simply incorrect, since metal clusters tend to fragment and fall apart when subjected to the intense electron beam dose that is used to generate images such as the STEM images shown in figure 2 d-g. This is one reason

where no ordered porosity is seen in Fig. 2 or in S8. Furthermore, drawing the circles around atoms with such low contrast (see fig. S8) is questionable with the amorphous background of the alumina showing enough random bright spots.

6) An example of over interpretation or imprecise characterization is on line 165 of page 7. What is 'fast' desorption of CO? When I study their FTIR spectra, I see the bands being fairly resistant to desorption in flowing N₂ at 30 C. The CO desorbs only after heating to 100 C. This is true of all of their spectra, which means the CO is quite strongly bound, unlike what they state in the text.

7) On line 171 page 7, they refer to 'embedded' morphology, another ill-defined term. How do they know the atoms are embedded and not on the surface? And if they were embedded, they might not be catalytically active. For this reason they need to report turnover frequency (TOF). They should base this per atom of Pt, since in this manner their catalyst reactivity can be compared with those of other workers. As it stands, they only show lightoff curves which depend on heat and mass transfer effects and do not represent kinetics. They need to report TOF at low conversions so they are free from mass and heat transfer limitations.

8) On line 197 page 8 they discuss how the XPS spectra indicate that the Pt species are mainly dispersed on the exterior domains of the γ -alumina. I fail to see how XPS can show this directly. The XPS samples a certain depth of their samples, depending on the energy of the x-rays used and the specific photoelectron being analyzed. And the severe overlap with the Al peaks makes the interpretation of the XPS (figure S 23 and S 24) questionable.

9) The EXAFS and XANES are convincing that their samples show absence of Pt-Pt coordination. But these are air exposed samples, where the Pt is exposed to air. I did not see any in-situ XPS demonstrating the nature of the working catalyst. For example, Figure 4 shows the EXAFS of the 0.2 m-alumina sample after CO oxidation, but I am not sure if the sample was exposed to air during transfer to the EXAFS?

10) Ultimately the evidence for the single atom nature of the 0.2 Pt/m-alumina lies in the CO FTIR which shows that the band position is different from that of 0.2 Pt/ γ -alumina. But it is a difference in degree, which means the same bands are present but in different proportions. So, what I see is that the authors have a sample that is kinetically trapped into a state where its high dispersion is preserved (due to the low loading). But this is a metastable state, since we see some larger clusters in this sample. This means that if the Pt atoms come into contact with each other, they will grow to form clusters. Which is why their sample retains its characteristics only at low loadings (as they admit in line 132 on page 5).

11) Despite the high dispersion of the Pt/m-alumina, the reactivity is comparable to the other catalysts. This is why they need to show TOFs to establish whether the reactivity is truly superior. For the other probe reaction, Fig. 5g, I noticed that the m-alumina catalyst is actually lower in reactivity than the commercial sample. And the high selectivity (Fig. 5a) is only seen at near zero conversion. The catalysts need to be compared at similar conversion.

Reference 45: the title of the paper, "eeposition" should be "deposition".

Reviewer #3 (Remarks to the Author):

The manuscript by Zhang et al. reports on the preparation and application of a new stable versatile single-atom Pt/ γ -Al₂O₃ catalyst for three representative reactions, under either oxidative or reducing gas mixtures at elevated temperatures up to 550 °C. The advantages of the catalyst were attributed to the clear single-atom feature of the platinum cations that were stabilized by the alumina lattice oxygen in a square-planar structure. Moreover, the embedded platinum atoms in the alumina substrate improved the thermal stability of the overall support texture. The work is a joint effort from multiple groups, and a formidable amount of raw/processed data has been included to show various properties of the new catalyst. Overall, the paper presents a compelling case for the stabilization of single Pt atoms in alumina to high temperatures, if the material is prepared as the authors demonstrate, and this is an important new piece of work in the area of single-atom supported metal catalysts. However, the data interpretation is confusing in several places throughout the paper. To meet the standards of Nature Communications, more work is needed to address in greater depth the (rather phenomenological) presentation of the experimental findings. A major revision with re-evaluation is thus recommended.

My detailed comments are as follows:

- 1) The materials investigated do not reflect the industrial significance advocated in the introduction. Indeed, the autocatalyst uses a lot of platinum every year as the authors cited, but NOT as Pt/Al₂O₃ catalysts. The formulation of Pt/Al₂O₃ has long been phased out from the mainstream three-way catalysts that are being used in conventional gasoline engine emission controls. Platinum is too volatile to survive the high-temperature aging, and palladium (major) and rhodium (minor) are the dominant precious metals being used. For diesel and other lean-burn gasoline engine emissions, platinum has been used widely, but again not as Pt/Al₂O₃ catalysts. The Pt-Pd alloy catalysts are now being widely used worldwide for this purpose. The alloy catalysts generally have a multifold better performance than the Pt-only catalysts for CO and HCs oxidation. Therefore, the catalyst development from the current work does not go into the heart of the technical challenge that we are facing today. The discussion in this section could be changed to reflect the practice more accurately.
- 2) According to the EXAFS results, it is claimed that the single-atom centric Pt-O₄ planar structure will fully survive the H₂ reduction at 400 °C, and that this is indeed the versatile catalytic center for various reactions. However, if one compares Fig. 3D with Fig. 4F, a clear decrease of the Pt-O peak intensity in the R-space of EXAFS data can be observed for the same “best” 0.2Pt/m-Al₂O₃-H₂ catalyst. Doesn't this indicate the evolution of the catalytic center? Along the same line, why does the “unreducible” Pt-O₄ planar structure in the 0.2Pt/m-Al₂O₃ catalyst give strong reduction peaks in H₂-TPR (Fig. S2)? What are the reducible species? Is adsorbed oxygen an issue even after the 50 °C- pretreatment in H₂? These experiments should be presented as 2nd or 3rd cycle, without exposure to oxygen between the cycles. If real, how do the reducible species relate to the catalytic activity? Why do the other higher Pt loaded Pt/m-Al₂O₃ catalyst (comprising clusters plus atoms) prepared by the same method give weaker reduction peaks? If the Pt-O coordination was intact before and after reduction, why do the 0.2Pt/m-Al₂O₃-H₂ and 0.2Pt/m-Al₂O₃-O₂ catalysts behave differently in CO oxidation? In Ref. 42, where the authors cited the DFT results to explain the reaction mechanism, the alumina support was found not the part of the CO oxidation catalysis. There is a critical inconsistency here, which the authors must address. The authors should also analyze the Pt edge in post-reaction catalyst by EXAFS for all

the reactions to confirm the stability and the coordination structure proposed according to the as-received sample.

- 3) For the CO oxidation reaction, a rather high contact time has been used (100 mg sample, 80 ml/min flow). Firstly, this does not well reflect the real application (suggest: 100 mg sample, 1-5 L/min flow). The more serious concern is that the key catalyst-- 0.2Pt/m-Al₂O₃ has minimal activity below 200 oC in repeated cycles, even at such a high contact time. This is the same issue encountered in Ref. 42. I did a calculation of turnover frequency (TOF) myself, and found that the TOF numbers from the present work and in Ref. 42 are surprisingly close. This reinforces the fact, from both papers, that the conventional Pt/Al₂O₃ catalyst with platinum particles and clusters present is seemingly more active for CO oxidation, even from the perspective of TOF per Pt atom. To deal with this issue, the authors simply cited the findings from Ref. 38, and claimed the particles are responsible for the low-temperature CO oxidation. This is convenient but unconvincing and controversial. If the Pt particles contain the low-temperature catalytic sites, how does the commercial Pt/Al₂O₃ with a lot of particles lose its low-temperature activity over the cycles? And why the 0.2Pt/p-Al₂O₃-H₂ sample that has (some and growing) clusters never became active for CO oxidation below 200 oC? Nonetheless, the single-atom Pt catalysts can be very active for low-temperature CO oxidation as a few earlier papers have pointed out [*Nat. Chem.* **3**, 634-641 (2011). *Angew. Chem. Int. Ed.* **53**, 8904-8907 (2014)]. However, one the conditions for activity are very different; e.g. in the former paper the PROX reaction is examined. Dry CO oxidation may be very different. What is the effect of water (-OH groups)? Is the catalyst rendered more active in the presence of water? This is an important question both from a fundamental and practical viewpoint that must be considered by the authors.
- 4) From the DFT results in Ref. 42, which the present work employs to offer mechanistic interpretations, it is reported that the highly active Pt₁-O_x structure only has two oxygen atoms from the substrate coordinating with the Pt directly. However, what the authors prepared experimentally in this paper is a very stable (meets 16 e rule) planar Pt-O₄ species. In the section of discussion, the author mentioned that “overly strong interaction leads to catalytically inactive species”. Is it possible that “over stabilization” applies to the authors’ own work? If the single-atom 0.2Pt/m-Al₂O₃ is not intrinsically more active for CO oxidation, what is the benefit of the new stabilized structures? Here the authors may invoke this stability as an attribute for some practical applications to be defined. Pertaining to autocatalysts, good Pt catalysts should have significant CO conversion below 200 oC—otherwise the subsequent HCs and NO oxidation will be greatly hindered.
- 5) The connection (or correlation) of the Pt ions to the Al(3+) penta-coordinated aluminum is left unanswered in the paper. Are these the binding sites for the Pt sites? (see ref. 34) Does the fact that ONLY 0.2 % Pt is found in isolated atom form relate to the number of these special Al sites? Can the Pt content increase to practical values (0.5-1.0 wt %) if the number density of these Al sites increases?

On p. 5 the authors state: In addition, the enhanced Pt content is detrimental to the formation of high quality hexagonally arranged mesoporous Al₂O₃ (Supplementary Fig. 10,11). This makes

me wonder if the method of preparation used here is really limited to very low Pt loadings. This is a drawback that must be made clear in the paper. On the discussion that follows still on p.8, another control sample for the authors to consider would be the addition of Pt by incipient wetness impregnation on their m-Al₂O₃. Can this Pt “anchor” on the special Al(3+) penta-coordinated sites? How much Pt can thus be stabilized? The m-Al₂O₃ has very high content (1/3) of these special sites..

- 6) How do the findings link to the classical debate of the structure sensitivity found in Pt/Al₂O₃ catalysts for CO oxidation? Are there two mechanisms for the reaction? One at low-temperature and one at high-temperature. How about kinetic measurements over the two different structures of Pt catalysts presented here? As for the Pt dispersion, measured here by H₂-O₂ titration and CO pulse chemisorption (never saturated?! Fig. S7), the authors do not have a table to summarize these results, and never use the standard term “dispersion” to describe these findings. I did a few calculations myself, and found that the relation between the kinetic rate and the total exposed Pt surfaces of the m-, p-, and commercial alumina supported samples does not have a clear trend. The authors need to ponder this important issue.
- 7) For the given CO oxidation and calcination temperatures up to 400 oC, I could not judge if the single-atom platinum is more stable than the alkali ion-stabilized single platinum atoms reported in *J. Am. Chem. Soc.* **137**, 3470-3473 (2015). To demonstrate the point that the single atoms in alumina prepared as shown here do have a superior stability, the main premise of this paper, the samples treated at 600 oC should be fully analyzed to show the exclusive presence of single atoms, and the activity should also be reported. The results for the samples treated at 800 oC are actually puzzling to me. The authors need to be aware that the platinum may have already vaporized at this temperature (see *Science* **353**, 150-154 (2016), and multiple other reports discussing Pt catalysts deactivation). How much Pt remain in the alumina after the 800 oC-treatment?
- 8) What is reason that leads to the incremental (not breakthrough) improvement for the hexane reactions? Again, the total amount of the exposed Pt surfaces in various samples that I estimated from the H₂-O₂ titration results could not explain the initial reaction rates. Does the chemical valence (metal vs. cation) matter to the hydrogenation and reforming reactions? Will the isolated metal Pt atom stabilized in an alternative substrate be a better solution? As in other places in the paper, the authors report good activity, but do not get to the heart of the chemistry involved. Potentially, the section on hexane can be removed, and published separately in a specialized journal. Already, the paper has too many figures, and parts for a Communications paper.
- 9) The stability of the m-Al₂O₃ seems to be poorer compared to many conventional alumina supports, although the Pt doping is found to stabilize the alumina framework. I suggest adding another reference sample by using the m-Al₂O₃ and loading the platinum by incipient wetness impregnation method. The commercial Pt/Al₂O₃ sample may have too many differences

besides the state of Pt to make fair comparisons with the in-house prepared Pt catalysts. For example, the unusual light-off curve shape for the commercial Pt/Al₂O₃ sample at high conversions may be due to the mass transfer related issues rather than the intrinsically modified chemistry.

- 10) "The strong metal support interaction" has its unique original meaning in catalysis beyond the expression of "the interaction between the metal and support is strong". The authors need to be cautious about this and modify their wording in the section of discussion.
- 11) Finally, the title of the paper should be more specific and spell out the Pt/m-Al₂O₃ being investigated. A more appropriate title would be: *Thermally stable single atom Pt/m-Al₂O₃ for CO oxidation and the selective hydrogenation of 1,3 butadiene*

Editing points:

1. p. 8, line 10 from the bottom, ref. should be 40, not 41.
2. P. 13, line 3 from the top, ref. 32 should be used instead of 45, 46 which are for non Pt metal.

Reviewers' comments:

Reviewer #1 (Remarks to the Author):

This is a very thorough paper describing the synthesis, characterization, reactivity and stability of high dispersion Pt catalysts on Al₂O₃. The developed synthetic protocol introduces high concentrations of Al³⁺ sites that act as strong binding sites for atomically dispersed Pt. It is shown that the samples containing high concentrations of Al³⁺ are able to maintain good stability for the atomically dispersed Pt under a wide range of conditions. The analysis and conclusions are for the most part convincing and publication in Nat Comm is recommended after a few issues are addressed.

1. It is argued that the Al³⁺ centers are critical binding sites for the Pt atoms. However, no Pt-Al interactions were observed or discussed on the XAS data. Some discussion on this should be added.

Response:

Penta-coordinated Al³⁺ on the surface has been proven to be the anchoring site for Pt ions (*Science*, **2009**, 325, 1670-1673). From high-resolution ²⁷Al solid state MAS-NMR study, the authors proposed that Pt atoms bind to the Al³⁺ penta sites on the γ-Al₂O₃ surface through oxygen bridges. This is in agreement with our observation in XAS analysis that Pt does not have direct bonding with Al but rather through oxygen linkages.

We have added discussions after presenting ²⁷Al NMR data concerning the anchoring mode of Pt on Al³⁺ centers (MS Page 8). ".....Considering there is no direct bonding between Al and Pt in EXAFS spectrum, Pt atoms plausibly bind to these sites via oxygen bridges³⁵. We further synthesized....."

2. The IR interpretation should be further substantiated.

a. For Pt/Al₂O₃, CO stretching frequencies below 2100 cm⁻¹ have been rigorously assigned to metallic adsorption sites on nanoparticle surfaces at either well coordinated (~2090 cm⁻¹) or undercoordinated (~2070 cm⁻¹) Pt sites (see *ACS Catal.* **2016**, 6, 5599–5609). It is surprising that CO adsorbed to the cationic Pt single atoms don't exhibit stretching frequencies > 2100 cm⁻¹. It is known that both Pt structure and oxidation state play a role in defining the CO stretching frequency, so this should be discussed and somehow rationalized. See *J. OF CATALYSIS* 169, 382–388 (1997) for a nice discussion on some of these issues.

Response:

We agree that many factors, including the electronic state, particle size, coordination environment, and metal-support interaction, will influence the CO adsorption characteristics. Thus, CO-DRIFT alone could not provide an unambiguous picture of the metal species on the support. The peak assignments varied from case to case. For example, 0.18 wt% Pt/ θ -Al₂O₃ single atom catalyst exhibits two CO adsorption peaks at room temperature: 2108 cm⁻¹ and 2056 cm⁻¹ (*J. Am. Chem. Soc.* **2013**, *135*, 12634–12645), the latter of which has been assigned as CO adsorbed at Pt(0) single-atom sites. This is different from both the example mentioned by the reviewer and our case.

In addition, CO adsorbed to positively charged Pt species does not necessarily exhibit stretching frequencies > 2100 cm⁻¹. Datye et al. assigned the absorption at 2095 cm⁻¹ to CO on cationic Pt(II) species, whereas CO adsorption peak on Pt₁^{δ+}/FeO_x (bader charge: +0.45e) located at around 2080 cm⁻¹. It appears the valent state of Pt₁ affects the CO peak position. The key tools for identifying single atom catalysts are EXAFS and AC-STEM. In our study, these techniques provided compelling evidence that the Pt single-atom species dominant in the sample.

Some discussions based on our observation and literature data (e.g., *ACS Catal.* **2016**, *6*, 5599-5609) have been added into the manuscript (MS Page 7, also see Response to 2c).

- b. The CO band observed in Figures 2F and 4E are missing some signatures of adsorption at single Pt sites. First, the band positions seem to shift with coverage, suggesting dipole interactions that shouldn't exist at single atom adsorption sites. Second, the bands are quite wide in FWHM and have some obvious asymmetry that suggests there are multiple adsorption sites being observed that have different chemical characteristics.

Response:

In literature, a shift of CO adsorption band as a function of coverage is well-known for nanoparticles. However, the shift is normally much larger than what we observed. For example, the shift was 7 cm⁻¹ on 2.6 wt% Pt/HZSM-5 (*Science*, **2015**, *350*, 189-1920) and 15 cm⁻¹ on 2.5 wt% Pt/FeO_x (*Nat. Chem.* **2011**, *3*, 634-641).

We propose the Pt₁ species in our catalyst are not identical, which may stay on slightly different coordination sites on the support. This leads to relatively wide FWHM and induces asymmetry to the CO adsorption peak. It also results in the shift of CO adsorption band when coverage changes.

We have added an auxiliary line to indicate the slight shift of CO adsorption peak as coverage decreases (see below), and the some text in the MS (see Response to c).

Revised **Figure 3f**.

- c. In the recent Science papers by Stair's group and Datye's groups CO was observed to stick very strongly for stretches that were assigned as single Pt atom adsorption sites. This does not agree with the results presented here. This should be discussed.

Response:

As discussed above, the CO adsorption behaviours of Pt catalysts are influenced by various factors, so that different Pt single-atom catalysts may have distinct CO adsorption strength. While the Pt₁ catalysts prepared by Stair's group and Datye's groups exhibited strong affinity to CO, our catalyst is similar to the Pt₁/FeO_x catalyst developed by Zhang's group, in which CO adsorption on catalysts is weaker. The CO adsorption behaviour on Pt₁/FeO_x catalyst is copied below, indicating CO readily desorbs when CO concentration decreases.

Figure 3a in *Nat. Chem.* **2011**, *3*, 634-641.

We thank the reviewer for providing excellent comments related to DRIFT-IR spectra. We have added further discussions and comparisons between our observation and literature data to the text “.....which can be rationally ascribed as linearly adsorbed CO on Pt^{39, 40}. For Pt nanoparticles supported on α -Al₂O₃, CO stretching frequencies below 2100 cm⁻¹ have been rigorously assigned to metallic adsorption sites on nanoparticle surfaces at either well-coordinated (~2090 cm⁻¹) or undercoordinated (~2070 cm⁻¹) Pt sites, but CO adsorbed on Pt^{δ+} single-atom sites may also exhibit a peak in this region. In fact, the adsorption strength and DRIFT-IR peak of CO on Pt₁ sites are highly dependent on the system. While Pt₁/ZSM-5 prepared by atomic layer deposition and Pt₁/CeO₂ prepared by atomic vapour trapping exhibited strong affinity with CO, with peak positioned at 2095-2115 cm⁻¹, Pt₁/FeO_x synthesized via sol-gel method showed weaker CO binding strength at 2080 cm⁻¹, which is similar to our catalyst. CO desorbed from 0.2Pt/m-Al₂O₃-H₂ at even room temperature, with 3 cm⁻¹ red shift observed, which reflects the atomically dispersed Pt species may have non-identical coordination environments.” (MS Page 7).

3. It is mentioned that the reactivity of the 0.2Pt-m/Al₂O₃ actually increases with time on stream. This is suggesting there may be some agglomeration of Pt atoms to form small metallic clusters that are more active for this reaction. Some suggestions of what structural or chemical changes may be occurring to the catalyst that induce this reactivity change should be added.

Response:

The slight activity enhancement of 0.2Pt/m-Al₂O₃ in CO oxidation with time on stream indeed suggests some structural changes of the active center. We have conducted XAS measurement and data fitting of spent catalyst after CO oxidation. There is no presence of Pt-Pt bond indicating negligible nanoparticle or nanocluster formation. In addition, the Pt-O coordination number and the white line intensity remain almost identical after reaction. On the other hand, we observed an increased Debye–Waller factor in the spent catalyst, reflecting some changes of Pt species during the reaction, despite Pt remains predominantly atomically dispersed.

We modified Figure 4f (MS Page 13) and added the fitting data of spent catalyst in Supplementary Table 4 (SI Page S54). We also added the following sentence to the MS to indicate possible changes of catalyst center based on XAS analysis: “.....In the EXAFS spectrum, Pt-O contribution located at approximately 1.7 Å remains as the only prominent shell, unarguably proving that Pt largely maintains single-atom identity (Fig. 4f, Supplementary Fig. 32e). While the white line intensity (1.64) and Pt-O coordination number (3.6) remain almost identical after reaction, an increase in Debye–

Waller factor was observed, reflecting some degree of evolution of catalyst active center (Supplementary Table 4).....” (MS Page 11).

Revised **Figure 4f**. k^3 -weighted Fourier transform spectrum derived from EXAFS for 0.2Pt/m-Al₂O₃-H₂ after the CO oxidation.

Reviewer #2 (Remarks to the Author):

This manuscript reports the synthesis, characterization and catalyst stability of Pt single-atom catalyst supported on mesoporous alumina. The manuscript is interesting in that the authors compare the performance of different alumina samples, exploring the role of alumina coordination on the nature of Pt species and their stability. This makes the work novel, since their synthesis involves commonly used reagents and alumina is among the most widely used catalyst supports, so the work could have broad applicability. However, the work does not represent any fundamental advance, since even transitional alumina was shown (ref 34) to stabilize single atoms. What the authors have demonstrated is that their catalyst is stable when the Pt concentration is low (0.2 wt%) preventing the Pt species from forming larger particles. At higher concentrations, the Pt tends to form larger particles. And it appears that the reactivity of these catalysts is not superior to that of the conventional catalysts. The work is not suitable for publication in its present form and needs major revisions to correct the errors in the interpretation of the data, as listed below. And the reactivity comparisons for CO oxidation need TOF data and for the selective hydrogenation, need to be compared at similar conversion.

Specific comments

1) The authors have synthesized mesoporous alumina (m-alumina), and then another one they call porous alumina (p-alumina). Since alumina tends to crystallize when heated, I would like to see evidence that the mesoporous alumina structure is preserved after their high temperature treatments. Supplementary figure 7 shows ordered structure in the as-prepared state. Figure S8 and S9 shows them after treatments, but the temperature is not stated, and the samples do not exhibit any order. Hence, the authors need to state whether or not the pore structure is retained. Otherwise, I would infer that the m- and p- aluminas are not probably very different after being heat treated.

Response:

The samples shown in Figure S7, S8 and S9 were 0.2Pt/m-Al₂O₃-O₂ and 0.2Pt/m-Al₂O₃-H₂, both of which were treated at 400 °C. We did not mention the temperature because “400 °C” is the “standard” treating temperature in this study, while any sample treated at a different temperature is clearly labelled. The HAADF-STEM images in Figure S8 and S9 were taken from ultra-thin areas at very high magnification to observe single Pt atoms, at which condition the ordered mesoporous structure of the support could not be seen clearly.

In fact, m-Al₂O₃ preserves its porous structure much better than p-Al₂O₃ in heating condition. From BET analysis (provided in Figure S26c,e), m-Al₂O₃ with 0.2 wt% Pt can maintain its surface area (> 250 m²/g) after 600 °C and 800 °C treatment. In sharp contrast, the BET surface area of 0.2 wt% Pt on p-Al₂O₃ decreased monotonically with increased treating temperature, dropping from 265 m²/g (treated at 400 °C) to 114 m²/g (treated at 600 °C), and further to 61 m²/g (treated at 800 °C). We also provided TEM images for the 0.2Pt/m-Al₂O₃-O₂ and 0.2Pt/p-Al₂O₃-O₂ after 600 and 800 °C treatment in Figure S27, further demonstrating the ordered structure of the 0.2Pt/m-Al₂O₃-O₂ was well preserved.

2) The N₂ adsorption isotherms in Fig. S5 do not suggest a broad pore size distribution, not what one would expect based on the TEM images (which of course sample very small regions of the specimen). Hence, the authors need to show the low angle XRD region which will show clearly the extent of ordered mesoporosity in their structure. This is important since the authors claim that their samples are stable under extreme conditions and the m-alumina is better, so establishing the stability of the alumina pore structure is important.

Response:

We have conducted small-angle XRD analysis on both 0.2Pt/m-Al₂O₃ catalyst (treated at 400 °C, 600 °C, and 800 °C, respectively) and 0.2Pt/p-Al₂O₃ control sample (treated at 400 °C), which provided strong evidence of the presence of hexagonally ordered mesopores (p6mm symmetry) for 0.2Pt/m-Al₂O₃ samples (Figure below). The XRD patterns show two reflections, (100) and (110), for the m-Al₂O₃ samples calcined at 400, 600, and 800 °C; the d-spacing values corresponding to (100) reflections are 11.1 nm for 0.2Pt/m-Al₂O₃-O₂-400, 10.0 nm for 0.2Pt/m-Al₂O₃-O₂-600, and 9.7 nm for 0.2Pt/m-Al₂O₃-O₂-800. This is in agreement with literature observation of mesoporous Al₂O₃ materials (*J. Am. Chem. Soc.* **2008**, *130*, 15210-15216). In sharp contrast, the XRD spectrum for 0.2Pt/p-Al₂O₃-O₂-400 is featureless, indicating the lack of ordered porosity. The small angle XRD patterns have been added as Supplementary Figure 5 and discussions have been provided to both MS (Page 4) and SI (Page S7).

Revised **Supplementary Figure 5** | Small-angle XRD patterns of 0.2Pt/m-Al₂O₃-O₂ and 0.2Pt/p-Al₂O₃-O₂ after high temperature treatment.

3) The manuscript should omit reporting data from the literature, for example figure S1 on utilization of Pt and Figure 1 in the manuscript, which is not reporting any original data from this study.

Response:

We have removed Figure 1a, 1b and Figure S1 in the revised MS and SI. A reference is given to Johnson Matthey's report reflecting the production and demand of Pt in recent years.

4) The interpretation and analysis of the images in Fig. 2 are incorrect. First, to image single atoms of Pt you need a probe diameter of sub-Angstrom size. Each bright dot will then correspond to the size of the probe. I cannot tell from the scale of the image if each of those dots meets this criterion. I do not understand the Gaussian fit of image g, which should show a single peak corresponding to the size of the probe (since the atom is much smaller in size). I cannot read the scale very clearly in this rather fuzzy image, but establishing this is important to convince the reader that they truly have isolated single atoms.

Response:

Yes, the observed STEM intensity is a convolution of the probe function and object function, and the probe size is a decisive factor for the image resolution. In this study, a probe of $\sim 1 \text{ \AA}$ was used and the FWHM of the Gaussian fitted peaks for single atoms are around $1.2 \sim 1.3 \text{ \AA}$, which is consistent with the value reported for single atoms in the literature (e.g. *Microsc. Microanal.* **2012**, *18*, 1342-1354). Slight variation in the FWHM among single atom peaks may result from beam-induced atomic motions that distort the contrast (*Microsc. Microanal.* **2010**, *16*(S2), 70-71) and/or a small deviation from the perfect focus. According to the reviewer's comments, we have made the scale of the profile clearer for an easy identification of the FWHM in the revised manuscript (see revised Figure 2h). Moreover, we use an atomic-resolution STEM image of Au nanoparticles taken under exactly the same imaging condition as a reference, in which the FWHM of profiles belonging to individual atomic columns are $1.4 \sim 1.5 \text{ \AA}$ (see the figure below as an example). These observations provide strong evidence that the bright dots in our HRSTEM images correspond to isolated Pt single atoms.

(left) HRSTEM image of Au nanoparticle and (right) Intensity profile and Gaussian fit of a single atomic column as shown by the red line in the left figure. The corresponding FWHM is $\sim 1.5 \text{ \AA}$.

5) Secondly, Fig. 2e is incorrect in asserting this to be the interparticle distance, since the STEM image in Figure 2 is a projection of the 3-D sample. Hence what you measure as the interatom distance is not really the true distance, it is a projection of this distance on to a plane. It is clear that the bright dots don't have the same contrast, so they are not in one plane. And to make any inferences from the interatomic distances in the cluster seen in figure 2g is simply incorrect, since metal clusters tend to fragment and fall apart when subjected to the intense electron beam dose that is used to generate images such as the STEM images shown in figure 2 d-g. This is one reason where no ordered porosity is seen in Fig. 2 or in S8. Furthermore, drawing the circles around atoms with such low contrast (see fig. S8) is questionable with the amorphous background of the alumina showing enough random bright spots.

Response:

We thank the reviewer for raising insightful comments. We totally agree that strictly speaking, the measured "inter-atom" distances here are projected distances.

We did not distinguish "projected distance" from the true, 3-D "inter-atom distances" in our original submission because of the following considerations: i) The HAADF-STEM images were taken from ultra-thin areas in the specimen to gain reasonable contrast of single atoms. Therefore, the difference in height between a single atom and its nearest neighbors is insignificant; and ii) HAADF-STEM with a large convergent angle (~ 27 mrad used here) exhibits high depth-sensitivity and this phenomenon is pronounced for single heavy atoms on a light support (*PNAS* **2006**, *103*, 3044-3048). The contrast of a single atom would decrease significantly upon a height change of only a few Ångstroms and completely blur out when larger height variation is introduced (*Appl. Phys. Lett.* **2005**, *87*, 034104). Therefore, the identified single atoms under a given focus condition should not differ much in height. Taken together, we thought that it would not bring much error to approximate the "projected distances" as "inter-atom distances". In the revised manuscript, we follow the reviewer's comments to make our statements more precise as follows. *Note that the distances between neighboring single atoms measured from HAADF-STEM are "projected distances". The true three-dimensional "inter-atom distances" should be larger. However, given that the HAADF-STEM images were taken from ultra-thin areas in the specimen to gain reasonable contrast of single atoms and that the image contrast is very sensitive to the vertical position of single atoms, the identified single atoms should not differ much in height. Therefore, it would not bring much error to approximate the "projected distances" as "inter-atom distances" in this specific case.* This paragraph of clarification has been provided in the caption of Supplementary Fig. 9, on page S10.

The reviewer is correct that most single atoms and small clusters exhibit considerable degree of structural dynamics upon the scanning of electron probe and that the cluster image in Fig. 2f ~ h should be more appropriately regarded as a “snapshot” of the dynamical structure. It is difficult to decide whether the observed loosely-packed cluster is an intrinsic feature or induced by the electron beam. According to this comment, we revised the corresponding statement as follows: *“Apart from the dominant amount of isolated atoms, a few small clusters that exhibit considerable structural dynamics under electron beam were also observed. A snapshot image (Fig. 2g, h) shows that the cluster has loosely-packed atoms with the interatomic distances longer than those observed in metallic Pt.³⁵ However, it is difficult to affirm whether these small clusters are formed by loose packing of single Pt atoms, or by electron beam-induced fragmentation of the close packed structure.”* (MS Page 4).

We thank the reviewer for the comments on the identification of single atoms. In this study, we identified Pt single atoms in the HAADF-STEM image by searching the local intensity maxima using the algorithm reported by I. F. Sbalzarini etc. (*J. Struct. Biol.* **2005**, *151*, 182-195). This method is not so sensitive to the contrast variation of substrates and works well for locating single atoms. The positions of single atoms are further refined and their coordinates are extracted for the nearest neighbor distance analysis. Specifically, we found that by using a aperture size of $\sim 1.6 \text{ \AA}$, a cut-off score of 0 for the non-particle discrimination, and an accepted percentile of 0.01 \sim 0.05, single atoms can be well identified in most HRSTEM images. Those located and refined single atoms are carefully inspected to avoid spurious identification before proceeding to the distance analysis. The detailed method for identifying single atoms from HAADF-STEM images has been added in the method section in SI (Page S3).

6) An example of over interpretation or imprecise characterization is on line 165 of page 7. What is ‘fast’ desorption of CO? When I study their FTIR spectra, I see the bands being fairly resistant to desorption in flowing N₂ at 30 C. The CO desorbs only after heating to 100 C. This is true of all of their spectra, which means the CO is quite strongly bound, unlike what they state in the text.

Response:

This is a nice comment. The CO adsorption on our catalyst is not particularly strong compared to some earlier Pt single-atom systems. For instance, no decrease of CO adsorption band was observed on 0.5Pt/HZSM-5 catalyst, even when the temperature increased to 100 °C for 15 min (*Science*, **2015**, *350*, 189-192). Similarly, there were no decay in CO adsorption band strength under Ar or O₂ flow at

50 °C on Pt₁/CeO₂ catalyst (Figure below, left, taking from *ACS Catal.* **2017**, *7*, 887-891) and only ca. 10% decay at 125 °C after 10 min on Pt₁/CeO₂ catalyst (Figure below, right, taking from *Science* **2016**, *353*, 150-154).

Nevertheless, we totally agree with the reviewer that the word “fast” does not provide a precise description of the CO desorption behaviour and should be avoided. We have modified the original statement as the following:

“The CO adsorption band dropped to ca. 1/3 of its original height after 30 min purging with N₂ at room temperature, and disappeared after heating at 100 °C for 2 min.” (MS Page 7)

7) On line 171 page 7, they refer to ‘embedded’ morphology, another ill-defined term. How do they know the atoms are embedded and not on the surface? And if they were embedded, they might not be catalytically active. For this reason they need to report turnover frequency (TOF). They should base this per atom of Pt, since in this manner their catalyst reactivity can be compared with those of other workers. As it stands, they only show lightoff curves which depend on heat and mass transfer effects and do not represent kinetics. They need to report TOF at low conversions so they are free from mass and heat transfer limitations.

Response:

We thank the reviewer for raising insightful comments. The word “embedded” has been removed in the revised manuscript.

We also agree that TOF is an important parameter to benchmark catalyst performances. CO oxidation over 0.2Pt/m-Al₂O₃-H₂ catalyst has been carefully measured between 180 and 250 °C, with 5 °C increment each step. The conversion of CO was kept low in the entire region to get rid of mass transfer limitations. TOF was 0.010 s⁻¹ at 180 °C and steadily increased to 0.17 s⁻¹ at 250 °C. There were five papers using Pt single-atom catalysts for CO oxidation, and our catalyst is comparable with

most values. We have compiled these TOF data and related catalyst/reaction information in Supplementary Table 6.

For selective hydrogenation of 1,3-butadiene, the TOF of our catalyst is 0.034 s^{-1} at $30 \text{ }^{\circ}\text{C}$. At 40 and $50 \text{ }^{\circ}\text{C}$, accurate TOF values could not be obtained due to high conversions. Nevertheless, we used the yield of butenes to estimate a low end of TOF at these temperatures ($>0.068 \text{ s}^{-1}$ at $40 \text{ }^{\circ}\text{C}$; $>0.12 \text{ s}^{-1}$ at $50 \text{ }^{\circ}\text{C}$). As we are aware, there is only one previous report using single Pt atom catalyst dispersed in Cu matrix for the reaction (*Nat. Commun.* **2015**, *6*, 8550). In that report, the TOF of Pt catalyst was 0.011 s^{-1} at $50 \text{ }^{\circ}\text{C}$, which is about an order of magnitude lower than our catalyst. The activity of Pt nanoparticles for 1,3-butadiene varied significantly in literature. Compared to these catalysts, $0.2\text{Pt}/\text{m-Al}_2\text{O}_3\text{-H}_2$ is either more active or much more selective. We have briefly discussed this in the main text and have included the comparison of TOF values and reaction selectivity of our catalyst and the ones from literature in Supplementary Table 10.

8) On line 197 page 8 they discuss how the XPS spectra indicate that the Pt species are mainly dispersed on the exterior domains of the γ -alumina. I fail to see how XPS can show this directly. The XPS samples a certain depth of their samples, depending on the energy of the x-rays used and the specific photoelectron being analyzed. And the severe overlap with the Al peaks makes the interpretation of the XPS (figure S 23 and S 24) questionable.

Response:

The reviewer has raised an excellent point. Indeed, the low Pt loading and the significant overlap with Al signal made XPS interpretation unconvincing. We have removed XPS spectra (original S23 and S24), experimental details and discussions in the revised MS and SI.

9) The EXAFS and XANES are convincing that their samples show absence of Pt-Pt coordination. But these are air exposed samples, where the Pt is exposed to air. I did not see any in-situ XPS demonstrating the nature of the working catalyst. For example, Figure 4 shows the EXAFS of the 0.2 m -alumina sample after CO oxidation, but I am not sure if the sample was exposed to air during transfer to the EXAFS?

Response:

XAS analysis was conducted in Japan so the sample was inevitably exposed to air during sample transfer and mailing. This may not be a big concern, however, due to the following reasons. First, the

CO oxidation reaction was conducted under O₂-rich atmosphere, so that the catalyst was always under oxidative conditions (the same as when exposed to air). Secondly, our catalyst is stable with O₂ in air. We have conducted CO-DRIFT study both *in-situ* and *ex-situ*, since the CO adsorption peak is sensitive to electronic and geometric characteristics of Pt species. For the *in-situ* measurement, the sample was pre-reduced with H₂ at 400 °C before CO adsorption (Figure 3f), whereas the *ex-situ* measurement was done with the sample transferred from the reactor to the DRIFT cell (Supplementary Figure 16a). The CO adsorption peaks were identical, suggesting exposure to air did not induce appreciable changes to Pt on m-Al₂O₃.

On the other hand, we fully agree with the reviewer that *in-situ* XAS technique would provide direct information concerning the key features of working catalyst. The collaboration between NUS team in Singapore and Kyoto U team Japan is continuing, and we have prepared and submitted a proposal to request beamtime for *in-situ* XAS analysis in the future.

10) Ultimately the evidence for the single atom nature of the 0.2 Pt/m-alumina lies in the CO FTIR which shows that the band position is different from that of 0.2 Pt/p-alumina. But it is a difference in degree, which means the same bands are present but in different proportions. So, what I see is that the authors have a sample that is kinetically trapped into a state where its high dispersion is preserved (due to the low loading). But this is a metastable state, since we see some larger clusters in this sample. This means that if the Pt atoms come into contact with each other, they will grow to form clusters. Which is why their sample retains its characteristics only at low loadings (as they admit in line 132 on page 5).

Response:

p-Al₂O₃ indeed provided a portion of Pt single-atoms in the fresh catalyst. However, its initial catalytic performance (both activity and selectivity) in all three probe-reactions were lower than Pt supported on m-Al₂O₃. Moreover, the long-term stability of Pt/p-Al₂O₃ were far worse than Pt/m-Al₂O₃ in *n*-hexane reforming reaction. These indicate that m-Al₂O₃ is unique in stabilizing Pt₁ species that survive harsh reaction conditions.

Most noble metal-based single-atom catalysts are metastable. Decrease metal loading is a common strategy to prevent metal atoms contacting each other. In most cases, however, single-atom species are not stable even if prepared at low loadings. For this reason, they were often tested under mild conditions. As we are aware, the stability of single-atom catalysts have seldom been scrutinized at temperature above 320 °C (*Nat. Chem.* **2011**, *3*, 634-641; *J. Am. Chem. Soc.* **2013**, *135*, 12634-12645;

Angew. Chem. **2014**, *126*, 9050-9053; *Science*, **2016**, *353*, 150-154; *ACS Catal.* **2017**, *7*, 887-891; et al.). Our catalyst fully maintained its activity in CO oxidation over a one-month period on stream with varying conditions, including 60 cycles between 100 and 400 °C, 220 h at 400 °C and 70 h at 230 °C. This is remarkable compared with a vast majority of earlier studies.

In the area of single-atom catalysis, the ultimate goal may not be synthesizing thermodynamically stable atoms, since these species are likely to be catalytically less active or inactive. A balance between stability and the catalytic activity is important. In our study, a system in which Pt single atom species exhibit high stability without comprising activity has been developed, and therefore holds potential to be expanded into a broad range of single-atom catalysts with improved stability.

11) Despite the high dispersion of the Pt/m-alumina, the reactivity is comparable to the other catalysts. This is why they need to show TOFs to establish whether the reactivity is truly superior. For the other probe reaction, Fig. 5g, I noticed that the m-alumina catalyst is actually lower in reactivity than the commercial sample. And the high selectivity (Fig. 5a) is only seen at near zero conversion. The catalysts need to be compared at similar conversion.

Response:

We agree with the first part of the comment. TOF values have been provided and compared with literature data in the revised submission (see Response to Comment 7).

Concerning the second part of the comment, there seems to be a misunderstanding from the reviewer on Fig. 5g. This figure was meant to provide a comparison of on-stream stability between 0.2Pt/m-Al₂O₃ catalyst and control samples. The x-axis refers to time on stream, whereas the y-axis refers to the rate of deactivation, measured by the drop of *n*-hexane conversion every hour. From reviewer's comment, we realized the original Fig. 5g is not reader friendly, and may be a bit misleading. We have replotted it (see below), using the ratio between activity at any time and initial activity as the y value, which highlights how fast and to which extent the catalysts loss activity under extreme conditions (550 °C, in the presence of H₂). From the new figure, it becomes easier to find that 0.2Pt/m-Al₂O₃ only lost 10% activity whereas control samples lost over 30% initial activity after 12 h reaction.

Revised Figure 5g: the catalyst deactivation at 550 °C.

12) Reference 45: the title of the paper, "eeosition" should be "deposition".

Response:

This error has been corrected.

Reviewer 3#:

The manuscript by Zhang et al. reports on the preparation and application of a new stable versatile single-atom Pt/ γ -Al₂O₃ catalyst for three representative reactions, under either oxidative or reducing gas mixtures at elevated temperatures up to 550 °C. The advantages of the catalyst were attributed to the clear single-atom feature of the platinum cations that were stabilized by the alumina lattice oxygen in a square-planar structure. Moreover, the embedded platinum atoms in the alumina substrate improved the thermal stability of the overall support texture. The work is a joint effort from multiple groups, and a formidable amount of raw/processed data has been included to show various properties of the new catalyst. Overall, the paper presents a compelling case for the stabilization of single Pt atoms in alumina to high temperatures, if the material is prepared as the authors demonstrate, and this is an important new piece of work in the area of single-atom supported metal catalysts. However, the data interpretation is confusing in several places throughout the paper. To meet the standards of Nature Communications, more work is needed to address in greater depth the (rather phenomenological) presentation of the experimental findings. A major revision with re-evaluation is thus recommended.

My detailed comments are as follows:

1) The materials investigated do not reflect the industrial significance advocated in the introduction. Indeed, the autocatalyst uses a lot of platinum every year as the authors cited, but NOT as Pt/Al₂O₃ catalysts. The formulation of Pt/Al₂O₃ has long been phased out from the mainstream three-way catalysts that are being used in conventional gasoline engine emission controls. Platinum is too volatile to survive the high-temperature aging, and palladium (major) and rhodium (minor) are the dominant precious metals being used. For diesel and other leanburn gasoline engine emissions, platinum has been used widely, but again not as Pt/Al₂O₃ catalysts. The Pt-Pd alloy catalysts are now being widely used worldwide for this purpose. The alloy catalysts generally have a multifold better performance than the Pt-only catalysts for CO and HCs oxidation. Therefore, the catalyst development from the current work does not go into the heart of the technical challenge that we are facing today. The discussion in this section could be changed to reflect the practice more accurately.

Response:

The reviewer is a real expert in catalysis. Indeed, the new catalyst developed in this paper does not directly tackle any immediate technical challenge in industry. We have removed all descriptions on catalytic converters in the introduction. We have also modified the section on CO oxidation (MS Page 10).

2-1) According to the EXAFS results, it is claimed that the single-atom centric Pt-O₄ planar structure will fully survive the H₂ reduction at 400 °C, and that this is indeed the versatile catalytic center for various reactions. However, if one compares Fig. 3D with Fig. 4F, a clear decrease of the Pt-O peak intensity in the R-space of EXAFS data can be observed for the same “best” 0.2Pt/m-Al₂O₃-H₂ catalyst. Doesn't this indicate the evolution of the catalytic center?

Response:

We sincerely thank the reviewer for raising such an insightful comment. After inspecting the data, we find a mistake in the data processing procedure for this particular sample. The sample was measured under fluorescence mode but the box “Natural log” was mistakenly ticked. Since " $mt = \ln(I_{\text{fluorescence}} / I_0)$ " was used instead of " $mt = I_{\text{fluorescence}} / I_0$ ", the EXAFS oscillation was attenuated, thus the Fourier transform was also attenuated. That is a major reason for the significant decrease of Pt-O peak from 6.8 (Figure 3d) to 4 (original Figure 4f).

After processing the appropriate data with proper parameter selection, we obtained a revised Figure 4f as shown below. Since the difference between the fresh catalyst (Pt-O height of 6.8) and spent catalyst (Pt-O height of 5.5) may still not be considered as insignificant, we further conducted curve fitting analysis. To our surprise, the Pt-O coordination number before and after CO oxidation are almost identical. The difference in Pt-O peak height is cancelled by the increase of the Debye-Waller factor. While it remains unclear where the increase comes from, the change of Debye-Waller factor reflects some evolutions of the catalyst center as the reviewer suggested. In fact, the slight enhancement of the catalytic activity during the first few cycles also points to the same direction (Supplementary Figure 34a).

We added the following discussion to the text “.....In the EXAFS spectrum, Pt-O contribution located at approximately 1.7 Å remains as the only prominent shell, unarguably proving that Pt largely maintains single-atom identity (Fig. 4f, Supplementary Fig. 32e). While the white line intensity (1.64) and Pt-O coordination number (3.6) remain almost identical after reaction, an increase in Debye-Waller factor was observed, reflecting some degree of evolution of catalyst active center (Supplementary Table 4).....” (Page 11).

Revised **Figure 4f** with proper parameters selected, and fitting curve included.

Revised **Supplementary Table 4**. EXAFS parameters of Pt foil, 0.2Pt/m-Al₂O₃-O₂, and 0.2Pt/m-Al₂O₃-H₂ and spent catalysts. C.N., coordination number; r, bond length; σ^2 , the Debye-Waller factor; ΔE_0 , inner potential correction to account for the difference in the inner potential between the sample and each FEFF simulated path. $\Delta k = 1.3 - 2.0^{*1,*2,*4}$; $1.7 - 3.2^{*3}$; $1.3 - 2.1^{*5}$; $1.4 - 3.4^{*6,*7}$

Samples	Shell	C.N.	r/Å	σ^2	ΔE_0 (eV)	White line intensity

0.2Pt/m-Al₂O₃-O₂ ^{*1}	Pt-O	3.8+/-1.8	2.023 +/-0.033	0.0053 +/-0.0044	1.6+/-5.7	1.66
0.2Pt/m-Al₂O₃-H₂ ^{*2}	Pt-O	4.0+/-1.2	2.009 +/-0.022	0.0055 +/-0.0029	-0.7+/-3.9	1.65
Pt foil ^{*3}	Pt-Pt	12 (fixed)	2.766 +/-0.002	0.0046 +/-0.0002	0.7+/-0.5	1.25
PtO₂ ^{*4}	Pt-O	6 (fixed)	2.016 +/-0.008	0.0028 +/-0.0005	0.0+/-1.6	2.20
0.2Pt/m-Al₂O₃-H₂ ^{*5} After CO oxidation	Pt-O	3.6+/-0.9	2.024 +/-0.019	0.0063 +/- 0.0026	6.6+/-3.3	1.63
0.2Pt/m-Al₂O₃-H₂ ^{*6} After reforming at 400 °C	Pt-O	1.3+/-0.4	2.056 +/-0.017	0.0024 +/-0.0022	8.1+/-3.7	1.46
	Pt-Pt	6.5+/-1.6	2.726 +/-0.015	0.0095 +/-0.0015	-6.6+/-3.3	
0.2Pt/m-Al₂O₃-H₂ ^{*7} After reforming at 550 °C	Pt-O	1.5+/-0.4	2.025 +/-0.015	0.0035 +/-0.0020	1.8+/-3.1	1.46
	Pt-Pt	5.6+/-1.0	2.732 +/-0.009	0.0076 +/-0.0010	-4.1+/-2.0	

2-2) Along the same line, why does the “unreducible” Pt-O4 planar structure in the 0.2Pt/m-Al₂O₃ catalyst give strong reduction peaks in H₂-TPR (Fig. S2)? What are the reducible species? Is adsorbed oxygen an issue even after the 50 °C- pretreatment in H₂? These experiments should be presented as 2nd or 3rd cycle, without exposure to oxygen between the cycles. If real, how do the reducible species relate to the catalytic activity? Why do the other higher Pt loaded Pt/m-Al₂O₃ catalyst (comprising clusters plus atoms) prepared by the same method give weaker reduction peaks? If the Pt-O coordination was intact before and after reduction, why do the 0.2Pt/m-Al₂O₃-H₂ and 0.2Pt/m-Al₂O₃-O₂ catalysts behave differently in CO oxidation? In Ref. 42, where the authors cited the DFT results to explain the reaction mechanism, the alumina support was found not the part of the CO oxidation catalysis. There is a critical inconsistency here, which the authors must address.

Response:

We conducted the 2nd and 3rd cycles of catalyst reduction, without exposure to air between cycles. Negligible amount of H₂ consumption was observed in these additional cycles. These new data were added into Supplementary Figure 1.

Revised **Supplementary Figure 1 | TPR profiles for Pt/m-Al₂O₃**. Reduction conditions: 5% H₂/N₂, 80 mL min⁻¹, 10 °C min⁻¹ heating rate (50–600 °C). Before heating, the samples were stabilized in 5% H₂/N₂ at 50 °C for 1 h.

When designing the TPR experiment, total amount of Pt was kept the same (0.5 mg) for all measurements, as we thought cationic Pt would be the main species consuming H₂. As such, 250 mg sample was used for 0.2Pt/m-Al₂O₃-O₂, 100 mg for 0.5Pt/m-Al₂O₃-O₂, and 25 mg for 2.0Pt/m-Al₂O₃-O₂. The H₂ consumption in 0.2Pt/m-Al₂O₃-O₂ was entirely induced by reducible species on the support, since Pt could not be reduced (as we later found out), while TPR signals for 0.5Pt/m-Al₂O₃-O₂ and 2.0Pt/m-Al₂O₃-O₂ were caused by both support and Pt. The TPR signal for 0.5Pt/m-Al₂O₃-O₂ was the highest because the much larger amount of sample used in analysis (2.5 times more than 0.5Pt/m-Al₂O₃-O₂, and 10 times more than 2.0Pt/m-Al₂O₃-O₂).

To interrogate the subtle composition difference between 0.2Pt/m-Al₂O₃-O₂ and 0.2Pt/m-Al₂O₃-H₂, we also sent them for elemental analysis. The results are shown below (added in Supplementary Table 2):

Sample	C (wt%)	H (wt%)	N (wt%)
0.2Pt/m-Al ₂ O ₃ -O ₂	0.79	2.31	0.97
0.2Pt/m-Al ₂ O ₃ -H ₂	Not detected	1.48	<0.50

The carbon, hydrogen and nitrogen content in the catalyst all decreased considerably after reduction. Based on these, we propose both the adsorbed surface oxygen species and the removal of C, N elements contribute to hydrogen consumption. The change of C, H and N contents in m-Al₂O₃

(presumably on support surface) after reduction is a possible origin for the slightly different catalytic behaviour of 0.2Pt/m-Al₂O₃-O₂ and 0.2Pt/m-Al₂O₃-H₂.

2-3) The authors should also analyze the Pt edge in post-reaction catalyst by EXAFS for all the reactions to confirm the stability and the coordination structure proposed according to the as-received sample.

Response:

We analysed the EXAFS of Pt catalyst after CO oxidation (see response to 2-1). Pt fully maintained single-atom identify despite of some structural evolutions. We also managed to conduct XAS experiments for spent catalyst after *n*-hexane reforming reaction at 400 and 550 °C. FT-EXAFS curves of the two samples are provided in Supplementary Figure 58, whereas the fitting results are added into Supplementary Table 4. Clearly, Pt species are no longer exclusively atomically dispersed, as indicated by considerable formation of Pt-Pt bonds. The white line intensities dropped as well. This is in agreement with CO-DRIFT analysis where a mixture of isolated atoms and nanoclusters/nanoparticles were identified.

We added the following discussion to the text “.....*The CO adsorption band broadened on 0.2Pt/m-Al₂O₃-H₂, with the main peak shifting from 2089 cm⁻¹ to 2084 cm⁻¹, together with a shoulder peak at 2060 cm⁻¹ (Supplementary Fig. 57). Meanwhile, there is a significant drop of Pt-O coordination number from 3.6 to 1.1 (after reaction at 400 °C) and 1.4 (after reaction at 550 °C), with concurrent increase of Pt-Pt coordination number to around 6 (Supplementary Fig. 58 and Supplementary Table 4). These suggested the formation of Pt⁰ nanoparticles/nanoclusters after reaction, but there is still a substantial amount of isolated Pt atoms in the catalyst...*” (Page 15-16).

Since the new XAS experiments were conducted in Spring-8 in Japan, we have modified the “Catalysts characterization” section and the “Acknowledgements” section to include the information (Page 18 and 22, respectively).

3-1) For the CO oxidation reaction, a rather high contact time has been used (100 mg sample, 80 ml/min flow). Firstly, this does not well reflect the real application (suggest: 100 mg sample, 1-5 L/min flow).

Response:

Our mass flow controller has a maximum capacity of 100 ml/L. To increase the GHSV by a factor of ten, we decreased the catalyst to 10 mg while maintain the total flow rate at 80 ml/min. The TOF values are comparable with those obtained at 100 mg catalyst loading (Supplementary Fig. 31a). Furthermore, the apparent activation energy and reaction order measurements (high temperature range, 235-280 °C) were also conducted at this loading, *vide infra*.

3-2) The more serious concern is that the key catalyst-- 0.2Pt/m-Al₂O₃ has minimal activity below 200 °C in repeated cycles, even at such a high contact time. This is the same issue encountered in Ref. 42. I did a calculation of turnover frequency (TOF) myself, and found that the TOF numbers from the present work and in Ref. 42 are surprisingly close. This reinforces the fact, from both papers, that the conventional Pt/Al₂O₃ catalyst with platinum particles and clusters present is seemingly more active for CO oxidation, even from the perspective of TOF per Pt atom. To deal with this issue, the authors simply cited the findings from Ref. 38, and claimed the particles are responsible for the low-temperature CO oxidation. This is convenient but unconvincing and controversial. If the Pt particles contain the low-temperature catalytic sites, how does the commercial Pt/Al₂O₃ with a lot of particles lose its low-temperature activity over the cycles? And why the 0.2Pt/p-Al₂O₃-H₂ sample that has (some and growing) clusters never became active for CO oxidation below 200 °C?

Response:

During paper revision, we have carefully measured the TOF of our catalyst between 180 and 280 °C, with 5 °C increment each step (Supplementary Figure 30). The conversion of CO was kept low in the entire region to get rid of mass transfer limitations. TOF was 0.023 s⁻¹ at 200 °C and steadily increased to 0.175 s⁻¹ at 250 °C. Compared with Ref 42 (which becomes Ref 43 in the revised paper), these values are indeed very similar (0.013 s⁻¹ at 200 °C and 0.187 s⁻¹ at 250 °C. we thank the reviewer for his/her sharpness and meticulousness!).

We believe commercial Pt/Al₂O₃ catalyst lose low temperature activity over the cycles due to size increase of Pt nanoparticles over time. The fresh catalyst contains Pt nanoparticles with size centered at 3.9 nm, whereas the average size increased to 9.1 nm after 10 cycles. Moreover, the size distribution became much broader after CO oxidation. The histogram of Pt nanoparticles before and after reaction have been added to Supplementary Figure 29e,f and 38g,h, respectively.

The exact reason why 0.2Pt/p-Al₂O₃-H₂ sample never became active for CO oxidation below 200 °C is not clear. CO oxidation is a structure sensitive reaction. Presumably, the percentage of

clusters/nanoparticles that are highly active for low temperature CO oxidation during the cycles, statistically, is always low during reaction on stream.

3-3) Nonetheless, the single-atom Pt catalysts can be very active for low-temperature CO oxidation as a few earlier papers have pointed out [*Nat. Chem.* **3**, 634-641 (2011). *Angew. Chem. Int. Ed.* **53**, 8904-8907 (2014)]. However, one the conditions for activity are very different; e.g. in the former paper the PROX reaction is examined. Dry CO oxidation may be very different. What is the effect of water (-OH groups)? Is the catalyst rendered more active in the presence of water? This is an important question both from a fundamental and practical viewpoint that must be considered by the authors.

Response:

We are aware of literature reports that water could promote CO oxidation activity by the formation of surface hydroxyl group, which subsequently modified the reaction pathway (e.g., *ACS Catal.* **2017**, *7*, 887-891). To identify whether such effect exists in our system, we conducted experiments to examine water effect during paper revision (10 mg catalyst mixed with 90 mg γ -Al₂O₃, flowing rate 80 mL/min). Water was introduced into the reactor by a syringe pump at an infusion rate of 44.1 μ L/h, equivalent to 50 mol% of CO in the flowing gas. No significant change of activity was observed. The TOF of the catalyst at 150, 200, and 250 °C were 0.012, 0.039, and 0.26 s⁻¹ without adding water, whereas these numbers became 0.015, 0.043, and 0.22 s⁻¹, respectively, in the presence of water. The effect of water on CO reactivity has been added into the text (MS Page 11).

4) From the DFT results in Ref. 42, which the present work employs to offer mechanistic interpretations, it is reported that the highly active Pt₁-O_x structure only has two oxygen atoms from the substrate coordinating with the Pt directly. However, what the authors prepared experimentally in this paper is a very stable (meets 16 e rule) planar Pt-O₄ species. In the section of discussion, the author mentioned that “overly strong interaction leads to catalytically inactive species”. Is it possible that “over stabilization” applies to the authors’ own work? If the single atom 0.2Pt/m-Al₂O₃ is not intrinsically more active for CO oxidation, what is the benefit of the new stabilized structures? Here the authors may invoke this stability as an attribute for some practical applications to be defined. Pertaining to autocatalysts, good Pt catalysts should have significant CO conversion below 200 °C—otherwise the subsequent HCs and NO oxidation will be greatly hindered.

Response:

Based on XAFS and XANES analysis, our Pt catalyst stay in a 16-e stable configuration with four oxygen atoms from the support. This only represents the rest state of Pt catalyst, and the structure of Pt maybe dynamic during catalytic cycles. Although the Pt catalyst is not highly active in dry CO oxidation, its hydrogenation activity is remarkable, indicating the stable 16-e structure does not significantly compromise catalytic reactivity. We speculate that the coordination between one or two surface oxygen and Pt is labile, and may be replaced by reagents during the catalytic cycle. As such, the Pt center always remain in a stable 4-coordinate, 16-e state. One of the best techniques to experimentally prove that is conducting *in-situ* XAFS analysis. The collaboration between NUS team in Singapore and Kyoto U team Japan is continuing, and we have prepared and submitted a proposal to request beamtime for *in-situ* XAS data in Spring-8.

As the reviewer pointed out, our catalyst is not suitable as an autocatalyst. Considering its excellent activity, selectivity and stability in hydrogenating a wide range of substrates, it may be used in reduction reactions under both mild and harsh conditions.

5) The connection (or correlation) of the Pt ions to the Al(3+) penta-coordinated aluminum is left unanswered in the paper. Are these the binding sites for the Pt sites? (see ref. 34) Does the fact that ONLY 0.2 % Pt is found in isolated atom form relate to the number of these special Al sites? Can the Pt content increase to practical values (0.5-1.0 wt %) if the number density of these Al sites increases? On p. 5 the authors state: In addition, the enhanced Pt content is detrimental to the formation of high quality hexagonally arranged mesoporous Al₂O₃ (Supplementary Fig. 10,11). This makes me wonder if the method of preparation used here is really limited to very low Pt loadings. This is a drawback that must be made clear in the paper. On the discussion that follows still on p.8, another control sample for the authors to consider would be the addition of Pt by incipient wetness impregnation on their m-Al₂O₃. Can this Pt “anchor” on the special Al(3+) pentacoordinated sites? How much Pt can thus be stabilized? The m-Al₂O₃ has very high content (1/3) of these special sites.

Response:

Penta-coordinated Al³⁺ on the surface has been proven to be the anchoring site for Pt ions (Ref 35), and we ascribe the high stability of Pt species to be associated with surface penta-coordinated Al³⁺. Nevertheless, there is a significant difference between our m-Al₂O₃ material and γ -Al₂O₃ used in Ref 35. In the latter case, the spin-lattice relaxation times of tetrahedral and octahedral alumina were almost identical (120 ms) whereas that for penta-coordinated Al³⁺ ions was much shorter (8 ms). Based on that, the authors concluded that most of tetrahedral and octahedral alumina locate in the

crystalline γ -Al₂O₃ framework whereas penta-coordinated Al³⁺ ions locate on the surface of the alumina support. In our system, the spin-lattice relaxation times for all three types of Al³⁺ species are similar (Supplementary Figure 22b), indicating a majority of penta-coordinated Al³⁺ ions stay in the bulk. 0.2 wt% may be related to the number of penta-coordinated Al³⁺ ions on the surface, which is only a small fraction. Based on this analysis, Pt content should be able to be increased to a higher value if the surface density of penta-coordinated Al³⁺ sites increases.

As we understand, the current system only work well with 0.2 wt% loading. When increased to 0.5 wt% loading, the major species were still Pt single-atoms but Pt-Pt bond became observable in XAFS analysis. It is a drawback considering practical applications. We have added the following statement in the final paragraph of the discussion “...An apparent limitation of the system is that the single-atom identify of Pt was achieved only at a low loading (0.2 wt%), while mixed active sites were observed at more practical Pt loadings...”

We have used pre-synthesized m-Al₂O₃ as support to prepare Pt catalyst via wet-impregnation method. A mixture of Pt single atoms and nanoparticles were obtained even at 0.2 wt% (Supplementary Figure 21), highlighting Pt precursor has to be introduced prior to m-Al₂O₃ formation. When Pt(IV) precursor is mixed with Al precursor and ethanol, it is reduced to Pt(II) (based on ESI-MS analysis) and plausibly coordinates with Al through oxygen linkages, which may be critical for the formation of four-coordinate, 16-e Pt structure during the formation of mesopores.

6) How do the findings link to the classical debate of the structure sensitivity found in Pt/Al₂O₃ catalysts for CO oxidation? Are there two mechanisms for the reaction? One at low-temperature and one at high-temperature. How about kinetic measurements over the two different structures of Pt catalysts presented here? As for the Pt dispersion, measured here by H₂-O₂ titration and CO pulse chemisorption (never saturated?! Fig. S7), the authors do not have a table to summarize these results, and never use the standard term “dispersion” to describe these findings. I did a few calculations myself, and found that the relation between the kinetic rate and the total exposed Pt surfaces of the m-, p-, and commercial alumina supported samples does not have a clear trend. The authors need to ponder this important issue.

Response:

We conducted measurements of apparent activation energy and reaction order at both lower temperature range (180-250 °C) at 100 mg catalyst loading and higher temperature range (235-280 °C) at 10 mg catalyst loading. CO conversion was maintained low in all cases. The apparent

activation energy for CO oxidation between 180-250 °C was 80.3 kJ/mol, whereas it was 77.5 kJ/mol between 235-280 °C. There does not seem to be a two-mechanism scenario in our system. CO and O₂ reaction orders have been measured at 195 and 250 °C. In both cases, O₂ displayed a positive order (+1.3-1.6) while CO displayed a negative order (ca. -0.7) suggesting a uniform reaction mechanism. These new data were presented in Supplementary Figure 30 and discussions were added to the main text (Page 10-11).

Revised **Supplementary Figure 30 | CO oxidation kinetics over 0.2Pt/m-Al₂O₃-H₂**. (a) TOF evolution with reaction temperature. (b) Arrhenius plots. (c) CO and (d) O₂ order measured at 195 °C over 100 mg 0.2Pt/m-Al₂O₃-H₂. (e) CO and (f) O₂ order measured at 250 °C over 10 mg 0.2Pt/m-Al₂O₃-H₂. Catalyst was diluted 10 times with pure Al₂O₃.

The dispersion and TOF values (CO oxidation) of 0.2Pt/m-Al₂O₃-H₂, Pt/p-Al₂O₃ and commercial Pt/Al₂O₃ were summarized in Supplementary Table 5, while comparisons with literature data were provided in Supplementary Table 6. TOF values for selective butadiene hydrogenation were added into Supplementary Table 10. Dispersion of Pt in 0.2Pt/m-Al₂O₃-H₂ based on H₂-O₂ titration exceeded 100%, possibly due to hydrogen spill over effect. This was also observed in Pt₁/θ-Al₂O₃ catalyst (*J. Am. Chem. Soc.* **2013**, *135*, 12634-12645).

7-1) For the given CO oxidation and calcination temperatures up to 400 °C, I could not judge if the single-atom platinum is more stable than the alkali ion-stabilized single platinum atoms reported in *J. Am. Chem. Soc.* **137**, 3470-3473 (2015). To demonstrate the point that the single atoms in alumina prepared as shown here do have a superior stability, the main premise of this paper, the samples treated at 600 °C should be fully analyzed to show the exclusive presence of single atoms, and the activity should also be reported.

Response:

The JACS paper mentioned by the reviewer is from Flytzani-Stephanopoulos's group, entitled "A Common Single-Site Pt(II)-O(OH)_x- Species Stabilized by Sodium on "Active" and "Inert" Supports Catalyzes the Water-Gas Shift Reaction". It presented an elegant study on water-gas shift reaction over alkali ion-stabilized single platinum atoms, not on CO oxidation reaction. It is not possible to conduct a direct comparison to our work due to different reaction systems. Nevertheless, we have inspected that paper to get some clues related to high temperature stability. In that study, the harshest reaction condition was to expose Pt catalyst at 400 °C, 350 °C and 300 °C for 2 h each, whereas we have maintained our catalyst working at the 400 °C for 220 h. Aggregation of Pt species into nanoparticles (~ 2 nm) was observed in their system (Figure S11 in the JACS paper) after reaction, but it did not happen to our catalyst based on CO-DRIFT and XAS study.

As we are aware, the stability of single-atom catalysts have seldom been scrutinized at temperature above 320 °C (*Nat. Chem.*, **2011**, *3*, 634-641; *J. Am. Chem. Soc.* **2013**, *135*, 12634-12645; *Angew. Chem.* **2014**, *126*, 9050-9053; *Science*, **2016**, *353*, 150-154; *ACS Catal.* **2017**, *7*, 887-891; et al.). Our catalyst fully maintained its activity over a one-month period on stream with varying conditions,

including 60 cycles between 100 and 400 °C, 220 h at 400 °C and 70 h at 230 °C. XAS and CO-DRIFT analysis provided strong evidence of the exclusive single-atom identity of Pt after long-term reaction.

The Pt species did not remain exclusively as single-atoms after treatment at 600 °C. Prompt by reviewer's request, we conducted 14 cycles of CO oxidation over this catalyst. It was very stable over the cycles. The reaction cycles have been added in the supplementary information as Figure 28e.

7-2) The results for the samples treated at 800 °C are actually puzzling to me. The authors need to be aware that the platinum may have already vaporized at this temperature (see *Science* **353**, 150-154 (2016), and multiple other reports discussing Pt catalysts deactivation). How much Pt remain in the alumina after the 800 °C-treatment?

Response:

We are aware of Pt evaporation at 800 °C in literature reports. Prompt by this comment, we measured the Pt content after thermal treatment at 600 °C and 800 °C by ICP-OES analysis. The Pt content was 0.21 wt% and 0.22 wt%, respectively. The slightly increased Pt content compared with samples treated at 400 °C may due to small weight decrease after heating at higher temperatures. These results were added as the last two entries in Supplementary Table 1.

8) What is reason that leads to the incremental (not breakthrough) improvement for the hexane reactions? Again, the total amount of the exposed Pt surfaces in various samples that I estimated from the H₂-O₂ titration results could not explain the initial reaction rates. Does the chemical valence (metal vs. cation) matter to the hydrogenation and reforming reactions? Will the isolated metal Pt atom stabilized in an alternative substrate be a better solution? As in other places in the paper, the authors report good activity, but do not get to the heart of the chemistry involved. Potentially, the section on hexane can be removed, and published separately in a specialized journal. Already, the paper has too many figures, and parts for a Communications paper.

Response:

All three catalysts (Pt on m-Al₂O₃, p-Al₂O₃ and commercial Al₂O₃) tested in *n*-hexane reforming underwent deactivation over time, but Pt/m-Al₂O₃ exhibited the least decrease of activity. We characterized the spent Pt/m-Al₂O₃ catalyst by XAFS analysis, confirming the existence of Pt-Pt bonds after both 400 °C and 550 °C reactions (Pt-Pt CN ≈ 6). Although Pt single-atom species on m-Al₂O₃ could not fully survive reforming reaction condition, m-Al₂O₃ seems to be able to prevent over-

agglomeration of Pt nanoparticles (CO-DRIFT analysis, Supplementary Figure 57) and carbon deposition (TEM and TGA analysis, Supplementary Figure 50 and 53), leading to improved catalytic performance over long run (despite incremental, as pointed out by the reviewer).

Due to the instability of isolated Pt species under reaction condition, the intrinsic activity of Pt(II) single-atom species compared with Pt(0) on nanoparticles in *n*-hexane reforming reaction remains unclear. It is very good advice to use a support having stronger anchoring effect with Pt ions to evaluate the intrinsic activity of isolated Pt atoms in reforming reaction and compare that with Pt nanoparticles, which, to our knowledge, has not done before, and we will investigate it in future work.

At present, we prefer to keep this part in the manuscript, as it tells the community the limit of the system, and points out a direction of future research.

9) The stability of the $m\text{-Al}_2\text{O}_3$ seems to be poorer compared to many conventional alumina supports, although the Pt doping is found to stabilize the alumina framework. I suggest adding another reference sample by using the $m\text{-Al}_2\text{O}_3$ and loading the platinum by incipient wetness impregnation method. The commercial Pt/ Al_2O_3 sample may have too many differences besides the state of Pt to make fair comparisons with the in-house prepared Pt catalysts. For example, the unusual light-off curve shape for the commercial Pt/ Al_2O_3 sample at high conversions may due to the mass transfer related issues rather than the intrinsically modified chemistry.

Response:

We have prepared another control sample as suggested by the reviewer, using $m\text{-Al}_2\text{O}_3$ as the support to load 0.2 wt% Pt by wetness impregnation method (denoted as 0.2Pt/ $m\text{-Al}_2\text{O}_3\text{-imp}$). A mixture of single-Pt species and Pt nanoparticles were generated (Supplementary Figure 21). When tested in CO oxidation reaction, its performance over 14 cycles lies between that of 0.2Pt/ $m\text{-Al}_2\text{O}_3\text{-H}_2$ and commercial Pt/ Al_2O_3 catalyst. I.e., the lower temperature activity (< 200 °C) quickly decreased over the first few cycles whereas higher temperature activity (> 220 °C) slightly increased.

10) “The strong metal support interaction” has its unique original meaning in catalysis beyond the expression of “the interaction between the metal and support is strong”. The authors need to be cautious about this and modify their wording in the section of discussion.

Response:

Thanks for pointing this out. Considering the unique meaning of “strong metal-support interaction (SMSI)” in catalysis, we have avoided the use of this expression in the revised manuscript.

11) Finally, the title of the paper should be more specific and spell out the Pt/m-Al₂O₃ being investigated. A more appropriate title would be: *Thermally stable single atom Pt/m-Al₂O₃ for CO oxidation and the selective hydrogenation of 1,3-butadiene*

Response:

We agree the title proposed by the reviewer sharpens the scope of the article. The title has been changed as suggested.

Editing points:

1. p. 8, line 10 from the bottom, ref. should be 40, not 41.
2. P. 13, line 3 from the top, ref. 32 should be used instead of 45, 46 which are for non Pt metal.

Response:

These errors have been corrected in the revised manuscript.

We highly appreciate the referees for their critical, insightful and very constructive comments, which helped us better understand and report the new catalyst.

Reviewers' comments:

Reviewer #1 (Remarks to the Author):

The authors nicely addressed all concerns. The paper is now recommended for publication.

Reviewer #2 (Remarks to the Author):

The authors have done an extensive effort to address successfully the majority of concerns raised by reviewers. Additions and clarifications to the main story have been made.

However, I have two areas of concern and one correction, which would be easy to address:

- The scale in Figure 2h still appears fuzzy. The author altered the image axes, but it still appears out of focus. Could they improve it.
- The histogram in Figure 2e does not appear to correlate with any of the images in Figure 2. The image that this data is coming from should be provided to give context to the image. As it stands, the chosen images in Figure 2 do not show the numbers of atoms with the nearest neighbor distances they indicate in the histogram. Where is the data that leads to this histogram?
- The abstract should be updated to reflect the fact that the m-Alumina samples did not maintain complete Pt atomic dispersion after high temperature treatment (Supplementary Figure 28), nor for n-hexane reforming (Supplementary Figure 57). The improvement over commercial catalyst is mentioned, but the loss of complete single atom dispersion is not mentioned in their abstract or conclusions."

Reviewer #3 (Remarks to the Author):

In the revised manuscript and their rebuttal, the authors have done a very good job in addressing the reviewers' comments in detail. I believe the paper, after minor further revision, can be accepted for publication in Nat. Communications.

About the revision, there are a few points I want to stress:

- 1) Pt-Ox configuration: square planar and sort of dynamic during reaction to allow for activity, according to the authors. While this configuration is stable, the activity is not high. Loading of platinum is limited to just 0.2 wt% to secure the isolated Pt-Ox configuration. This low loading may be due to the small # of pentahedral Al (3+) on the surface. Authors state they cannot titrate the surface amount of these special aluminum sites because most of the signal is from the bulk ones which are not relevant. I agree. What do they propose as a good approach to the general reader?
- 2) The 0.2 wt.% Pt single atom catalyst that was heated to 600 and 800 o C shows good cyclic performance up to 400 o C in subsequent CO oxidation reaction. The new figure S28 is convincing. Fig. S27 shows NP formation after the high-temperature treatment and retention of some of the single Pt sites as well. The activity plot resembles that of samples heated to lower temperatures before the reaction. How are these findings reconciled?
- 3) The new Fig. S1 shows much higher reduction from the m-alumina sample. There may be H₂O desorption from this special (hydrothermally prepared, mesoporous) alumina, which has nothing to do with H₂O produced by H₂-TPR. Need to check this point by simply running desorption in He.
- 4) All the Pt samples (best and references) in this paper seemed to have darker color than the pure m-alumina (Fig. S3). I am not fully convinced that the best catalyst is Pt NP-free and the reference is simply too poor.

5) The response 3-2 is not really convincing. Moreover, in the related response 3-6, the kinetics show that the single-atom Pt-O4 is behaving essentially the same as conventional Pt/Al₂O₃ NP catalysts in both activation energy and rxn orders. If they are catalytically the same, why are the single atoms inferior in low-temperature CO oxidation activity? In their original submission, the authors claimed there are two stages of kinetics for NPs and single atoms, which have now been disputed by their own kinetics. Table S6 does not answer the question either— it simply indicates that their atomic catalyst is as inactive as others. Otherwise, there must be a significant presence of Pt metal that hides the presence of cationic Pt in kinetics.

For the final version of the paper, the authors should put the emphasis on the hydrogenation reaction, where the new catalyst is exciting and most promising; and limit the interpretation of CO oxidation data as a way of examining stability in oxidative atmosphere— not activity.

Throughout the paper, in IR parts, please use CO absorption band—not adsorption band.

Reviewers' comments:

Reviewer #1 (Remarks to the Author):

The authors nicely addressed all concerns. The paper is now recommended for publication.

Response:

No further action is required.

Reviewer #2 (Remarks to the Author):

The authors have done an extensive effort to address successfully the majority of concerns raised by reviewers. Additions and clarifications to the main story have been made.

However, I have two areas of concern and one correction, which would be easy to address:

- The scale in Figure 2h still appears fuzzy. The author altered the image axes, but it still appears out of focus. Could they improve it.

Response:

Thanks for the comment. Following the reviewer's suggestion, we have replaced the current profiles with those directly exported, high-resolution profiles (300 dpi) in the new version (Figure 2h on page 6).

- The histogram in Figure 2e does not appear to correlate with any of the images in Figure 2. The image that this data is coming from should be provided to give context to the image. As it stands, the chosen images in Figure 2 do not show the numbers of atoms with the nearest neighbor distances they indicate in the histogram. Where is the data that leads to this histogram?

Response:

Thanks for reviewer's comment. The statistical analysis of the nearest neighbor distances is carried out based on multiple HRSTEM images with larger field of view. We show representative HRSTEM images for each catalyst (i.e. 0.2Pt/m-Al₂O₃-O₂ & 0.2Pt/m-Al₂O₃-H₂) as examples in Supplementary Figure 8. This has been mentioned in Figure 2 footnote in the revised MS.

- The abstract should be updated to reflect the fact that the m-Alumina samples did not maintain complete Pt atomic dispersion after high temperature treatment (Supplementary Figure 28), nor for n-hexane reforming (Supplementary Figure 57). The improvement over commercial catalyst is mentioned, but the loss of complete single atom dispersion is not mentioned in their abstract or conclusions."

Response:

Thanks for the comments. The fact that the Pt/m-Al₂O₃ catalyst did not fully maintain single-atom identify was reflected in the revised abstract. "...this system also exhibited significantly enhanced stability and performance for n-hexane hydro-reforming at 550 °C for 48 h, **although agglomeration of Pt single-atoms into clusters was observed after reaction.**"

Reviewer #3 (Remarks to the Author):

In the revised manuscript and their rebuttal, the authors have done a very good job in addressing the reviewers' comments in detail. I believe the paper, after minor further revision, can be accepted for publication in Nat. Communications.

About the revision, there are a few points I want to stress:

1) Pt-Ox configuration: square planar and sort of dynamic during reaction to allow for activity, according to the authors. While this configuration is stable, the activity is not high. Loading of platinum is limited to just 0.2 wt% to secure the isolated Pt-Ox configuration. This low loading may be due to the small # of pentahedral Al (3+) on the surface. Authors state they cannot titrate the surface amount of these special aluminum sites because most of the signal is from the bulk ones which are not relevant. I agree. What do they propose as a good approach to the general reader?

Response:

Thanks for reviewer's comment. We have titrated the percentage of surface Al^{IV}, Al^V and Al^{VI} species based on literature reports that these surface species are interchangeable, i.e., the surface Al^{IV} and Al^V species interact with adsorbed water molecules to be transformed to Al^{VI} species via partial hydrolysis, whereas Al^{VI} species could be converted back to Al^V and Al^{IV} species under high temperature treatment (*J. Am. Chem. Soc.* **1998**, *120*, 11419; *J. Am. Chem. Soc.* **2000**, *122*, 12842; *J. Phys. Chem. C*, **2015**, *119*, 3428). We prepared a new batch of 0.2Pt/m-Al₂O₃-H₂ catalyst. The sample was loaded immediately into a NMR rotor in a glove box after reduction at 400 °C, before solid-state ²⁷Al NMR spectrum was collected (sample denoted as "dry"). Afterwards, we exposed the same sample in air for a few hours, transferred it back to the rotor, and measured the spectrum again using the same parameters (sample denoted as "after exposure to air"). A substantial change of the relative intensities of the peaks was observed. Based on curve fitting, the percentage of Al^V remain largely unchanged, while 7% Al^{IV} was converted into Al^{VI} species. Considering the surface area to be 200-260 m²/g, the exposed surface Al species is 5-7 % in m-Al₂O₃. This number matches the percentage of Al^{IV} transformed to Al^{VI} species under exposure to air. As such, the new NMR experiment suggests the dominant surface species is Al^{IV} for freshly prepared catalyst, which easily converts into Al^{VI} species in air.

The low abundance of Al^V on the surface may be the major reason why m-Al₂O₃ only stabilize atomically dispersed Pt atoms up to 0.2 wt%. Future studies should be directed to the enhancement of surface percentage of Al^V species. We have added the new NMR spectra as Supplementary Figure 22 f-g, and discussions in MS (page 8 & 17). We hope the message is clear to general readers (as well as ourselves).

	Octahedral Al	Pentahedral Al	Tetrahedral Al
Dry	14.6%	53.4%	32.0%
After exposure to air	21.8%	53.6%	24.6%

2) The 0.2 wt.% Pt single atom catalyst that was heated to 600 and 800 °C shows good cyclic performance up to 400 °C in subsequent CO oxidation reaction. The new figure S28 is convincing. Fig. S27 shows NP formation after the high-temperature treatment and retention of some of the single Pt sites as well. The activity plot resembles that of samples heated to lower temperatures before the reaction. How are these findings reconciled?

Response:

We have plotted the cyclic performance of 0.2Pt/m-Al₂O₃-400 and 0.2Pt/m-Al₂O₃-600 in the same figure (see below), taking data from 1st, 5th and 10th cycles, respectively. Indeed, the two catalysts exhibited almost identical activities (again, we are impressed by the reviewer’s sharpness and meticulousness!). This could be rationalized by the fact that Pt single-atoms are still dominant in 0.2Pt/m-Al₂O₃-600, which is consistent with CO-DRIFT observations (Figure S28). Pt nanoclusters and nanoparticles that are present in 0.2Pt/m-Al₂O₃-600 may have higher CO oxidation activity. However, these species account for a small percentage of Pt in the sample. Furthermore, their relative higher activity may be canceled by the smaller dispersion of Pt atoms.

3) The new Fig. S1 shows much higher reduction from the m-alumina sample. There may be H₂O desorption from this special (hydrothermally prepared, mesoporous) alumina), which has nothing to do with H₂O produced by H₂-TPR. Need to check this point by simply running desorption in He.

Response:

Thanks for reviewer's comment. We considered the possibility of H₂-TPR signal complexation by adsorbed H₂O on Al₂O₃. Therefore, the samples were treated at 150 °C under N₂ for one hour before cooling down to 50 °C to start the H₂-TPR experiments. This experimental detail has been added into the SI (page S4). To confirm whether this treatment is sufficient to remove adsorbed water in Al₂O₃, we have further conducted TPD experiments for 0.2Pt/m-Al₂O₃ in He, both with and without pre-heating under He at 150 °C for one hour. A peak was present over the sample without pre-heating but the peak disappeared after pre-heating at 150 °C. This suggests our pre-heating step is sufficient to remove most of adsorbed H₂O species on Al₂O₃. The new TPD profiles have been plotted and added as Figure S1b.

4) All the Pt samples (best and references) in this paper seemed to have darker color than the pure m-alumina (Fig. S3). I am not fully convinced that the best catalyst is Pt NP-free and the reference is simply too poor.

Response:

Thanks for reviewer's comment. We agree that the color of the material, to some extent, is an indicator of the existence of Pt NPs. Nevertheless, the single-atom Pt is dominant in the 0.2 wt% Pt samples, while the portion of Pt NPs is below the detection limit of XAS analysis. We have added the discussion following Supplementary Figure 3.

5) The response 3-2 is not really convincing. Moreover, in the related response 3-6, the kinetics show that the single-atom Pt-O4 is behaving essentially the same as conventional Pt/Al₂O₃ NP catalysts in both activation energy and rxn orders. If they are catalytically the same, why are the single atoms inferior in low-temperature CO oxidation activity? In their original submission, the authors claimed there are two stages of kinetics for NPs and single atoms, which have now been disputed by their own kinetics. Table S6 does not answer the question either— it simply indicates that their atomic catalyst is as inactive as others. Otherwise, there must be a significant presence of Pt metal that hides the presence of cationic Pt in kinetics. For the final version of the paper, the authors should put the emphasis on the hydrogenation reaction, where the new catalyst is exciting and most promising; and limit the interpretation of CO oxidation data as a way of examining stability in oxidative atmosphere— not activity.

Response:

Thanks for the excellent point. We do not totally understand the catalytic behavior of the single-atom Pt catalyst in CO oxidation. What is certain from the experiments is that the 0.2 wt% Pt/m-Al₂O₃ catalyst is not very active but stable under the reaction condition. Similarly, we could not fully rule out the

possibility of activity contribution from a small portion of highly active NPs for CO oxidation. Due to these considerations, we have adopted the reviewer's suggestion to limit the interpretation of CO oxidation data as a way of examining STABILITY in oxidative atmosphere, while we have strengthened discussions on selective hydrogenation with more discussions provided. Consequently, the title of the paper has been changed as "Thermally stable single atom Pt/m-Al₂O₃ for selective hydrogenation and CO oxidation", and the abstract has been modified.

6) Throughout the paper, in IR parts, please use CO absorption band—not adsorption band.

Response:

Thanks for pointing out the misspelling. Corrections have been made in the revised version.

REVIEWERS' COMMENTS:

Comments made to the editor.